# The nuclear pore primes recombination-dependent DNA synthesis at arrested forks by promoting SUMO removal

Karol Kramarz [1,2,3], Kamila Schirmeisen [1,2,3], Virginie Boucherit[1,2,3], Anissia Ait Saada [1,2,3], Claire Lovo [1,2], Benoit Palancade [4], Catherine Freudenreich [5] & Sarah A. E. Lambert [1,2,3 ✉]

Nuclear Pore complexes (NPCs) act as docking sites to anchor particular DNA lesions facilitating DNA repair by elusive mechanisms. Using replication fork barriers in fission yeast, we report that relocation of arrested forks to NPCs occurred after Rad51 loading and its enzymatic activity. The E3 SUMO ligase Pli1 acts at arrested forks to safeguard integrity of nascent strands and generates poly-SUMOylation which promote relocation to NPCs but impede the resumption of DNA synthesis by homologous recombination (HR). Anchorage to NPCs allows SUMO removal by the SENP SUMO protease Ulp1 and the proteasome, promoting timely resumption of DNA synthesis. Preventing Pli1-mediated SUMO chains was sufficient to bypass the need for anchorage to NPCs and the inhibitory effect of poly-SUMOylation on HR-mediated DNA synthesis. Our work establishes a novel spatial control of Recombination-Dependent Replication (RDR) at a unique sequence that is distinct from mechanisms engaged at collapsed-forks and breaks within repeated sequences.

---

[1] Institut Curie, PSL Research University, UMR3348, F-91405 Orsay, France. [2] CNRS UMR3348 "Genome integrity, RNA and Cancer", "Equipe labellisée LIGUE 2020", F-91405 Orsay, France. [3] University Paris Sud, Paris-Saclay University, UMR3348, F-91405 Orsay, France. [4] Université de Paris, CNRS, Institut Jacques Monod, F-75006 Paris, France. [5] Department of Biology, Tufts University, Medford, MA 02155, USA. ✉email: sarah.lambert@curie.fr

Flaws in the DNA replication process, known as replication stress, lead to fragile replication fork structures prone to chromosomal rearrangement and mutation, contributing to human diseases including cancer[1,2]. The resolution of replication stress occurs within a compartmentalized nucleus. How the distinct nuclear compartments operate to ensure faithful resolution of replication stress is far from understood.

The completion of DNA replication is continuously threatened by numerous obstacles. Replication obstacles hinder fork elongation and occasionally cause dysfunctional forks, deprived of their replication competence[3]. Replication-based pathways have evolved to ensure DNA replication completion and avoid genome instability. Dysfunctional forks are either rescued by opposite forks or, if a converging fork is not available in a timely manner, restarted and repaired. Homologous recombination (HR) is a ubiquitous DNA repair pathway involved in the repair of double strand breaks (DSBs), and in the protection and restart of dysfunctional forks[3]. This last pathway is referred to as recombination-dependent replication (RDR), a DSB-free mechanism allowing efficient fork-restart. The pivotal HR protein is the recombinase Rad51 that is loaded onto single-stranded DNA (ssDNA) with the help of its loader Rad52 in yeast. At compromised forks, the combined action of nucleases promotes the resection of newly replicated strands to generate ssDNA gaps and the subsequent loading of Rad51[4]. Then, the strand exchange activity of Rad51 builds a particular DNA structure, called a D-loop, from which DNA synthesis is primed allowing fork-restart[5,6]. A feature of RDR is its mutagenic DNA synthesis prone to chromosomal rearrangements[7-10]. How the subsequent steps of RDR are spatially segregated within the nuclear architecture is unknown.

The nuclear periphery (NP) constitutes a boundary between the nucleus and cytoplasm and is formed of a double membrane nuclear envelope (NE) and multiple nuclear pore complexes (NPCs)[11]. NPCs are highly conserved macromolecular structures, composed of multiple copies of 30 different nucleoporins, most of which associate in stable sub-complexes[12-14]. A central channel (referred to as the core of NPCs) allows macromolecule exchange between the cytoplasm and the nucleus. The largest NPC sub-complex is the Y-shaped mammalian Nup107-Nup160 complex (called Nup84 complex in budding yeast), located both at the cytoplasmic and nuclear side[15].

In budding yeast, DNA lesions (persistent DSBs, eroded telomeres, and collapsed forks) shift to the NP to associate with two distinct perinuclear anchorage sites: either the inner nuclear membrane SUN protein Mps3 or NPCs (extensively reviewed in ref. [16]). DSB-NPC association occurs in all cell cycle phases whereas DSB-Mps3 association is restricted to S/G2 cells. Relocation of DSBs to either Mps3 or the NPC requires distinct signaling mechanisms to promote distinct DNA damage survival pathways[17-24]. The fission yeast homologue of Mps3, Sad1, was shown to co-localize with DSBs, indicating an evolutionarily conserved role of the NE in DSB repair[25].

Anchoring of DNA lesions to NPCs requires SUMOylation events, a type of post-translational modification[17,20,22,23,26]. The SUMO (Small Ubiquitin-like Modifier) particle is covalently bound to lysines of target proteins by the joint action of SUMO-activating (E1) and -conjugating (E2) enzymes, a process enhanced by SUMO E3 ligases[27,28]. Persistent DNA damage and eroded telomeres are subject to SUMOylation waves that target DNA repair factors[29,30]. SUMOylated proteins are key substrates for the SUMO Targeted Ubiquitin Ligase (STUbL) family of E3 ubiquitin ligases such as the yeast Slx8-Slx5 and human RNF4, that target DNA lesions to NPCs[17,20,22,23,26,31-33]. SUMOylated proteins can undergo degradation or direct SUMO removal by SENP proteases, which are spatially segregated within the nucleus[34]. In yeasts, the SENP protease Ulp1 is constitutively attached to NPCs, whereas Ulp2 is found in the nucleoplasm[35,36].

The NPC has emerged as a central player in the maintenance of genome integrity[37,38]. Mutations in the budding yeast Nup84 complex lead to a defective DNA repair and replication stress response[11,17,36,39-41]. The outcome of relocation of damage is often deduced from the phenotypes arising from the ablation of anchorage sites at NPCs. Budding yeast NPCs favor the repair of DSBs by Break Induced Replication (BIR)[20,42]. Eroded telomeres relocate to NPCs in a SUMO-dependent manner to allow recombination-mediated elongation of telomeres, generating type II survivors[23]. A failure in anchoring forks stalled at expanded CAG repeats leads to chromosomal fragility of CAG tracts[22]. Also, delocalization of Ulp1 caused by mutations in the Nup84 complex results in DNA damage sensitivity[36] but how Ulp1-associated NPCs safeguard genome integrity is poorly understood. In eukaryotes, breaks within repeated sequences (Heterochromatin, rDNA) shift away from their chromatin environment, in a SUMO-dependent manner, to allow Rad51 loading and the completion of HR repair[26,43-46]. Thus, an emerging scenario suggests that NPCs are involved in both SUMO homeostasis and anchoring of DNA lesions to spatially segregate DNA repair events and avoid inappropriate HR repair. However, failures in uncoupling SUMO homeostasis from anchorage did not allow interrogating the relative contributions of these two NPC functions in maintaining genome integrity.

Using a site-specific replication fork barrier (RFB), we report that DSB-free and dysfunctional forks relocate and anchor to NPCs, in a poly-SUMO and STUbL-dependent manner, for the time necessary to complete RDR. Relocation occurs after Rad51 binding and enzymatic activity, suggesting that D-loop intermediates anchor to NPCs. We reveal a novel post-anchoring function of NPCs in promoting the removal of SUMO chains by Ulp1 and the proteasome. Indeed, the E3 SUMO ligase Pli1 safeguards fork-integrity and generates SUMO chains that trigger NPC anchorage but further limit the efficiency of HR-mediated DNA synthesis. Selectively preventing Pli1-dependent SUMO chains is sufficient to bypass the need for NPC anchorage in promoting HR-mediated DNA synthesis. We uncovered a novel SUMO-based regulation that spatially segregates the subsequent steps of RDR and that is distinct from mechanisms engaged at DSBs and collapsed forks within repeated sequences.

## Results

To investigate the spatial regulation of RDR, we exploited the *RTS1*-RFB that allows a single replisome to be blocked in a polar manner at a defined locus on *S. pombe* chromosome III (Fig. 1a). The RFB activity is mediated by the *RTS1*-bound protein Rtf1 whose expression is repressed in the presence of thiamine[47]. Forks arrested at the RFB become dysfunctional and are rescued by opposite forks or, if not available in a timely manner, restarted; both pathways require the binding of Rad51 to the active RFB[6]. Replication fork restart occurs by RDR within ~20 min and is initiated by an end-resection machinery to generate ssDNA gaps onto which RPA, Rad52, and Rad51 are loaded[4,5,48,49]. RDR is associated with a non-processive DNA synthesis liable to replication slippage and GCRs, during which both strands are synthetized by Polymerase delta, making the progression of restarted forks likely insensitive to the RFB[7,49].

### Dysfunctional forks associate with NPCs for ~20 min during S-phase. To follow the sub-nuclear location of the active RFB in living cells, we employed a *LacO*-marked RFB visualized by *LacO*-bound mCherry-LacI foci in yeast expressing the endogenous tagged Npp106-GFP, a component of the inner ring complex of

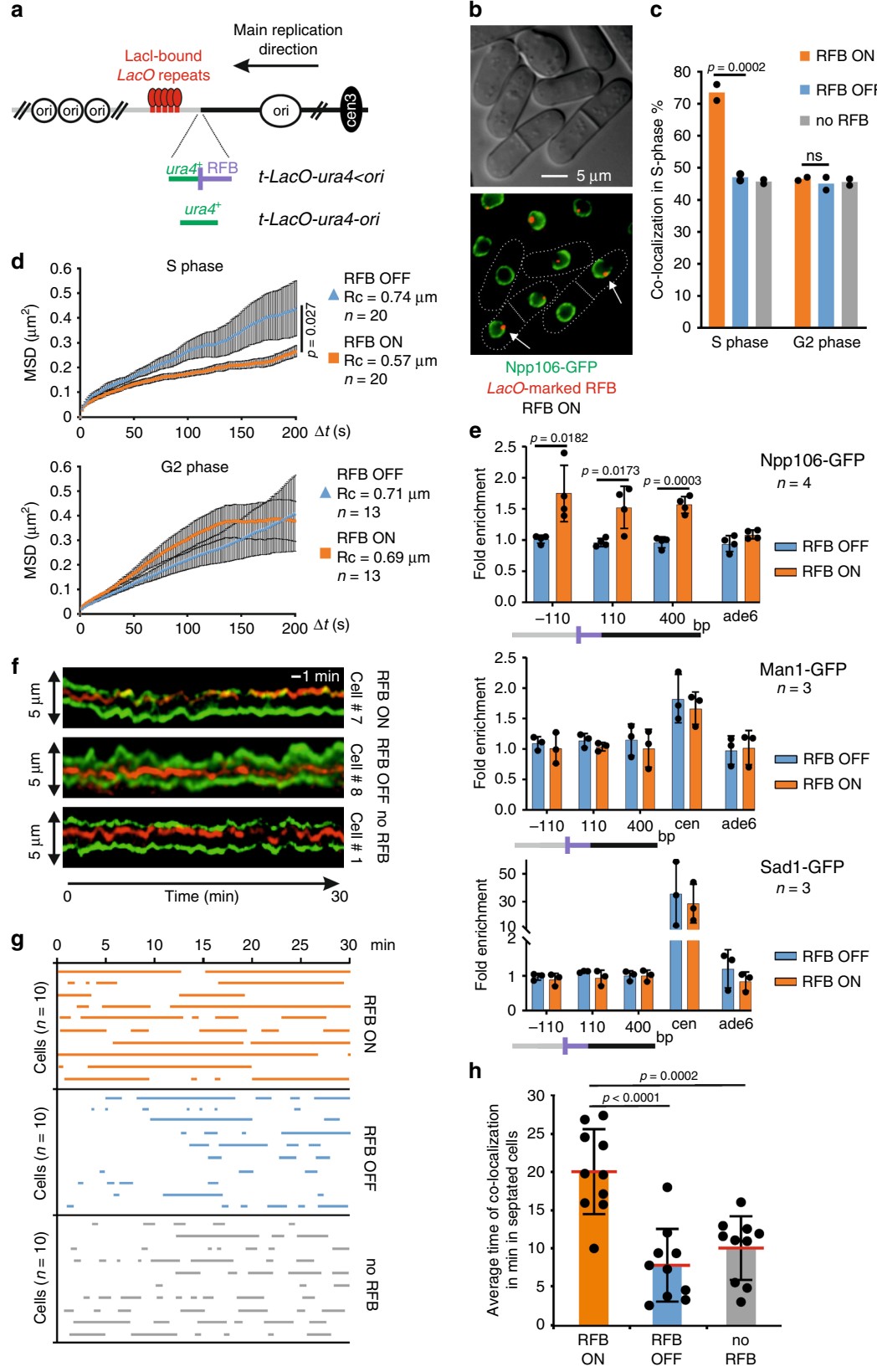

NPCs (Fig. 1a, b)[6]. The shape of the nucleus in S and G2-phase cells was often irregular, preventing us to apply a classical zoning approach[17] to assign the nuclear positioning of the LacO-marked RFB. Instead, we monitored co-localization between the NP and the LacO-marked RFB (Fig. 1b, c). When the RFB was inactive

(RFB OFF) or absent from the $ura4^+$ locus (no RFB, Fig. 1a), LacI-foci co-localized with the NP in ~45% of both S and G2-phase cells (Fig. 1c). Upon activation of the RFB (RFB ON), the LacO-marked RFB was located more frequently at the NP in S-phase cells, ~70% of the time, but not in G2 cells. Thus, forks

**Fig. 1 The active *RTS1*-RFB transiently relocates to NPCs in S-phase. a** Scheme of the *LacO*-marked *RTS1*-RFB (purple) integrated at the *ura4+* locus (green, *t-LacO-ura4 < ori*) or not (*t-LacO-ura4-ori*). Cen3: centromere position. *LacO* arrays (red) bound by mCherry-LacI (ellipses) are integrated ~7 kb away from *ura4+*. When Rtf1 is expressed (RFB ON, 24 h induction for cell imaging experiments) and binds to *RTS1*, 90% of forks moving from *cen3* to *t* are blocked. **b** Example of co-localization between Npp106-GFP and the *LacO*-marked RFB. Mono-nucleated cells and septated bi-nucleated cells correspond to G2 and S-phase cells, respectively. Arrows indicate co-localization events. **c** Quantification of co-localization events in indicated conditions: *t-LacO-ura4-ori*, Rtf1 expressed (no RFB), *t-LacO-ura4 < ori*, Rtf1 repressed (RFB OFF) and *t-LacO-ura4 < ori*, Rtf1 expressed (RFB ON). $n = 250$ cells in both S and G2 phase. Two-sided Fisher's exact test was used for group comparison to determine the $p$ value (ns non-significant). Dots represent values from two independent biological experiments. **d** The mobility of the RFB in OFF and ON conditions is presented as a mean square displacement (MSD) over the indicated time interval ($\Delta t$) for n independent cells. $R_c$ radius of constraint. $p$ value was calculated as a one sided $t$-test based on MSD curves. Black bars correspond to standard error of the mean (SEM). **e** Binding of the RFB to Npp106-GFP (top), Man1-GFP (middle) and Sad1-GFP (bottom) analyzed by ChIP-qPCR. Distances from the RFB are presented in bp. A centromere locus, known to interact with Man1 and Sad1 was used as a positive control. Primers targeting *ade6* gene were used as unrelated control locus. Values are mean of n independent biological repeats, with standard deviation (SD) as error bars. $p$ value was calculated using two-sided $t$-test. **f** Representative kymographs over 30 min of single S phase nucleus in indicated conditions. Green and red signals correspond to the Npp106-GFP marked nuclear periphery and the *LacO*-marked RFB, respectively. **g** Co-localization time from the analysis of kymographs in indicated conditions. Each line corresponds to an individual S-phase nucleus. Ten cells per conditions were analyzed. **h** Average co-localization time obtained from **f**. Each dot represents one sample, red bar indicate the mean from 10 independent S-phase cells ± SD. $p$ value was calculated using two-sided $t$-test.

arrested by a DNA-bound protein complex transiently relocate to the NP in S-phase cells.

To examine if the dynamics of the active RFB changes with NP enrichment, we monitored the mobility of the GFP-LacI focus by single-particle tracking (SPT) in living cells (Supplementary Fig. 1a) and calculated the range of nuclear volume explored by the *LacO*-marked RFB by mean square displacement (MSD) analysis (Fig. 1d) as reported for other types of damage[50]. Upon RFB activation, the overall mobility of the *LacO*-marked RFB decreased, exclusively in S-phase cells, compared to the RFB OFF control. The radius of constraint (Rc, radius of maximum volume of particle movement) in the OFF condition was significantly higher than the one obtained in the ON condition in S phase cells ($p < 0.05$) while no significant difference was detected in G2 cells, indicating that dysfunctional forks exhibit a reduced mobility in S-phase, consistent with an anchorage to a perinuclear structure. To identify the anchorage site, we performed Chromatin Immunoprecipitation (ChIP) experiments against Npp106-GFP, Sad1-GFP (the Mps3 orthologue) and Man1-GFP (a Lap-Emerin-Man domain protein of the inner nuclear envelope) to test their binding to the RFB. Man1 and Sad1 were found enriched at centromeres, as reported[51,52], but not at the active RFB (Fig. 1e). Npp106-GFP was significantly enriched at the active RFB, indicating that NPCs are acting as anchorage sites as reported for extended CAG repeats[22]. In these experiments, we used strains devoid of the nearby *LacO* array to ensure the binding of NP components to the active RFB is not a consequence of proximal *LacO* arrays that may influence sub-nuclear positioning.

To investigate the dynamics of the association of the RFB with the NP in single cell, we performed time-lapse microscopy for 30 min to build up kymographs over time (See "Methods" and Supplementary Fig. 1a). The analysis of 10 individual S-phase nuclei showed short and intermittent co-localizations between the NP and the unstressed locus (RFB OFF and no RFB controls), indicating transient and dynamic interactions (Fig. 1f, g and Supplementary Fig. 1b–d). The average time of co-localization was ~10 min (Fig. 1h). Consistent with an anchorage to NPCs, the active RFB co-localized with the NP in a less sporadic manner, with interactions lasting for most of the acquisition time in the majority of S-phase cells analyzed. The average time of co-localization was ~20 min (Fig. 1h), and correlated with the time needed to restart replication forks[48,49]. We conclude that dysfunctional forks transiently anchor to NPCs in S-phase, for a time that coincides with the time needed to complete RDR.

**Relocation to NPCs requires Rad51 loading and enzymatic activity.** Collapsed forks but not stalled forks associate to

NPCs[17,22]. Because the exact nature of DNA structures underlying collapsed versus stalled forks remains debated, we addressed the role of fork processing in anchoring the RFB to NPCs. The resection of nascent strands at arrested forks primes RDR. It occurs as a two-step process: a short-range resection by MRN-Ctp1 that generates ~110 bp sized gaps obligatory for replication restart followed by an Exo1-mediated long-range resection[5]. One role of MRN-Ctp1 is to remove the heterodimer KU from dysfunctional forks to overcome its anti-resection activity. Consequently, the lack of KU results in extensive fork-resection. We observed a lack of correlation between the extent of fork-resection and the capacity of the active RFB to shift to the NP and bind to NPCs (Fig. 2a, b, see Supplementary Fig. 2 for location in G2-phase). Instead, we noticed that RFB relocation was abrogated in mutants exhibiting a delay in replication restart (*i.e. rad50Δ*, *ctp1Δ* and *pku70*[5]) raising the possibility that replication/recombination intermediates formed during RDR trigger relocation to NPCs. Consistent with this, Rad51 and Rad52 were necessary to shift the active RFB to the NP (Fig. 2c and Supplementary Fig. 2). Rad51 promotes replication restart at arrested forks and protects them from uncontrolled end-resection to facilitate merging with opposite forks. To distinguish between these two Rad51 functions, we analyzed the *rad51-II3A* mutant that binds DNA to protect forks but is unable to facilitate restart because of its defective strand exchange activity[6]. The active RFB did not shift to the NP nor bind to NPCs in *rad51-II3A* cells (Fig. 2b, c and Supplementary Fig. 2), reinforcing the notion that relocation occurs after fork remodeling by Rad51 enzymatic activity. Since MRN-Ctp1 is active in *rad51-II3A* cells, we propose that short-range resection mediated by MRN-Ctp1 is necessary but not sufficient to shift arrested forks to NPCs and that building Rad51-mediated joint-molecules at arrested forks is necessary for stable association with NPCs.

**RDR and anchorage, but not fork-integrity, are impaired by the loss of the Slx5-Slx8 STUbL pathway.** Depending on the nature of DNA lesions, the *S. pombe* Slx8 STUbL either suppresses or promotes genome instability[53]. Also, Slx8 prevents uncontrolled HR at the constitutive *RTS1*-RFB[54]. Thus, it was worthwhile to address the role of SUMO and Slx8 activity in the spatial regulation of RDR. SUMO (encoded by the non-essential *S. pombe* gene *pmt3+*) was necessary to shift the active RFB to the NP in S-phase (Fig. 3a and Supplementary Fig. 3a). In the temperature-sensitive *slx8-29* mutated strain[55], the active RFB did not shift to the NP at 32 °C (Fig. 3a and Supplementary Fig. 3a) and MSD analysis showed an

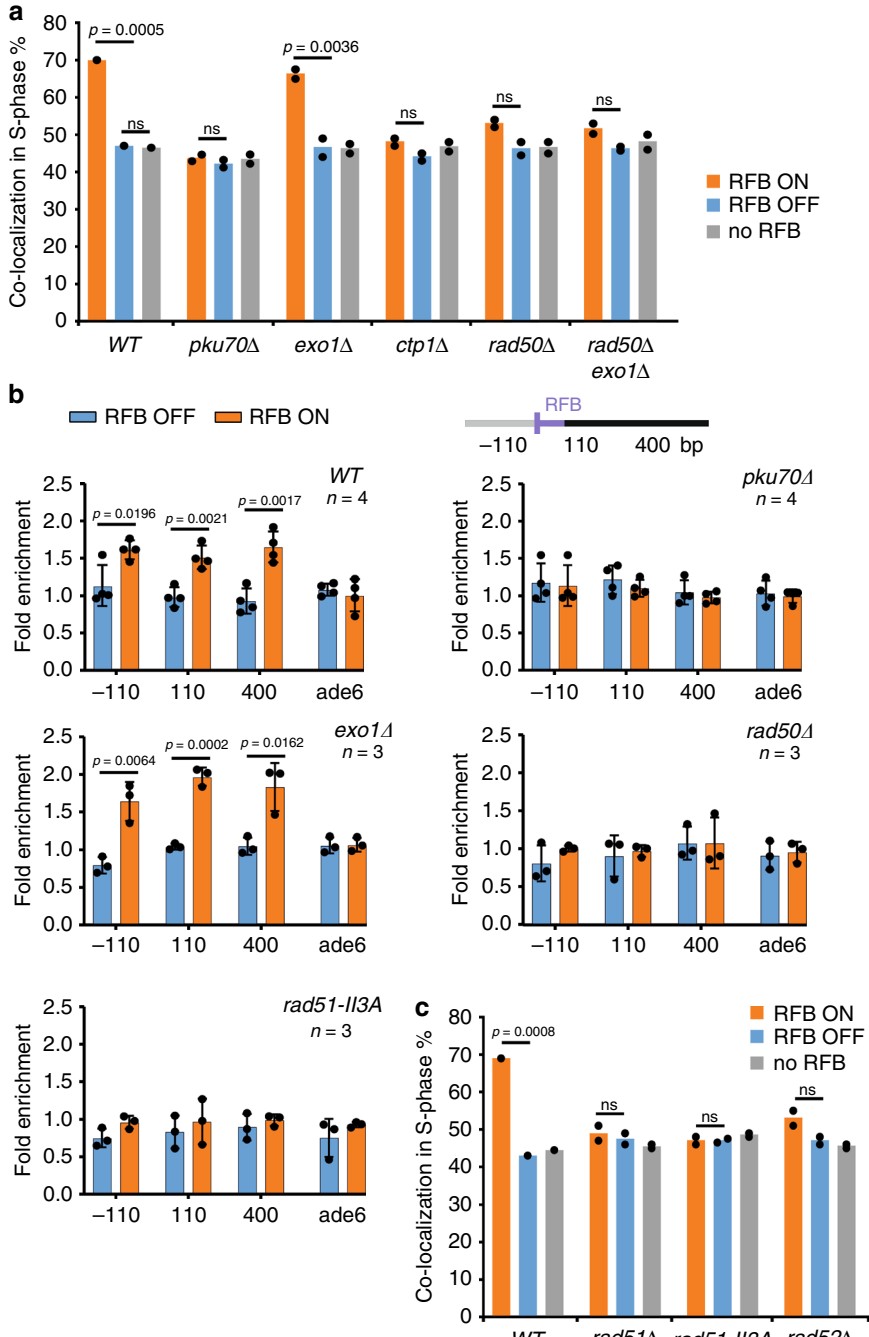

**Fig. 2 Relocation to NPCs requires Rad51 enzymatic activity. a** Co-localization events in S-phase cells in indicated conditions and strains, as described on Fig. 1b, c. *p* value was calculated by Fisher's exact test for OFF and ON groups for each mutant and condition. In all, 200 cells were analyzed for each strain and condition. Dots represent values obtained from two independent biological experiments. For each set of data, *WT* strain was analyzed alongside mutants. **b** Binding of Npp106-GFP to the RFB in indicated strains. Upstream and downstream distances from the RFB are presented in bp (top). Primers targeting *ade6* gene were used as unrelated control locus. Values are mean of n independent biological repeats, with SD as error bars. *p* value was calculated using two-sided *t*-test. **c** Co-localization events in S-phase cells in indicated conditions and strains, as in **a**.

increased mobility of the active RFB (Fig. 3b), indicating a lack of anchorage to NPCs when Slx8 is not functional. At permissive temperature (25 °C), the *slx8-29* mutated strain behaved as *WT* control (Figs. 3b and 1d). Rfp1 and Rfp2 are two orthologues of Slx5 and they form two independent heterodimers with Slx8[31]. The active RFB did not shift to the NP in the absence of either Rfp1 or Rfp2 (Fig. 3a and Supplementary Fig. 3a), reinforcing the notion that the Slx8 STUbL anchors arrested forks to NPCs.

To address the consequences of this lack of relocation, we investigated the efficiency of RDR. HR-mediated fork restart is associated with a non-processive DNA synthesis liable to replication slippage (RS). We developed genetic assays to monitor RFB-induced RS, based on the restoration of a functional *ura4+* gene to select for Ura+ cells (Fig. 3c and details in the legend)[7]. The frequency of Ura+ reversion is used as readout of the frequency at which the *ura4-sd20* allele is replicated by a restarted fork in the cell population. At 32 °C, the frequency of

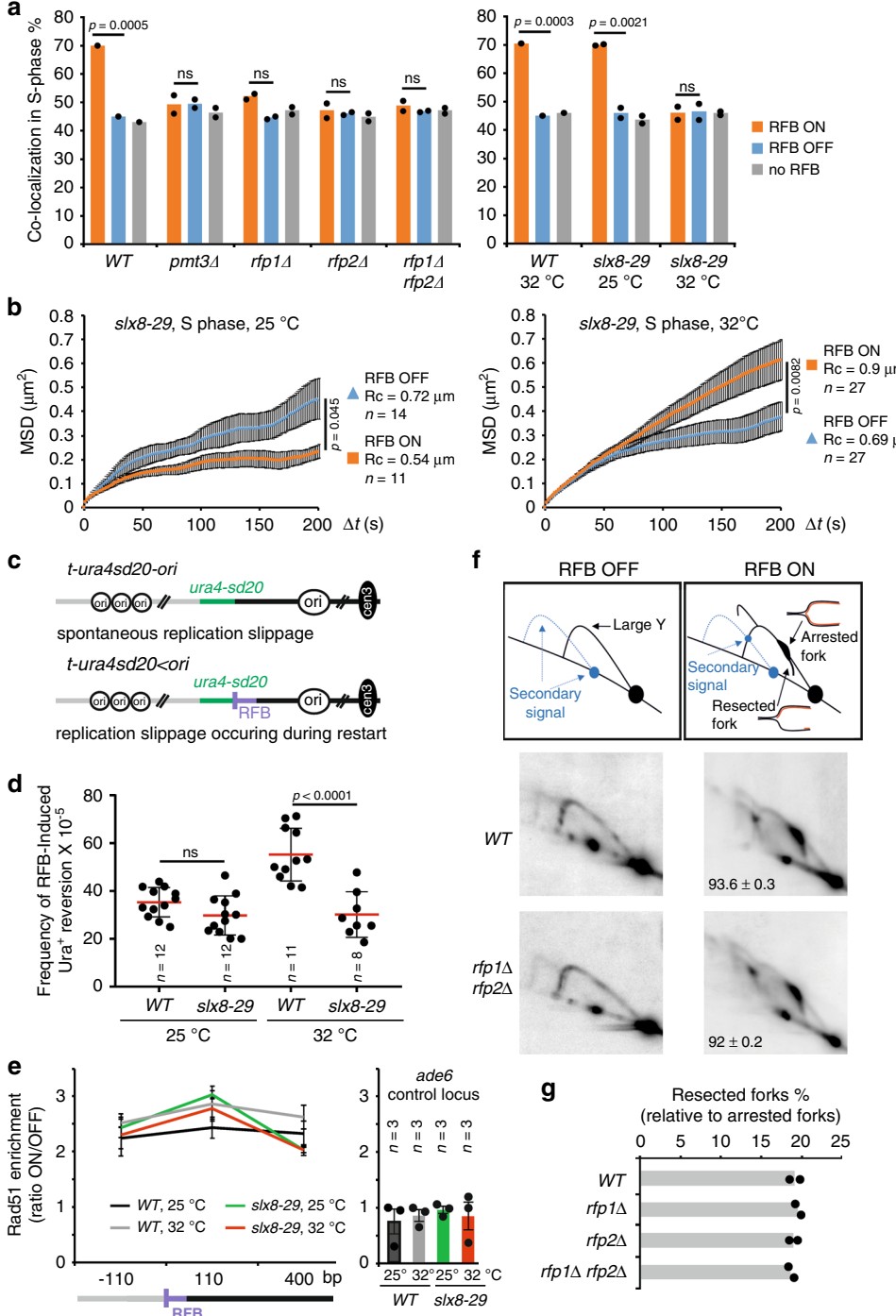

RFB-induced RS in *slx8-29* cells was decreased by nearly 50%, compared to *WT* (Fig. 3d) indicating that Slx8 promotes RDR. This defect was not caused by a less efficient Rad51 binding to the active RFB (Fig. 3e). Finally, we investigated the integrity of fork arrested by the RFB. We analyzed replication intermediates by bi-dimensional gel electrophoresis (2DGE) to examine the resection of nascent strands at arrested forks (referred to as resected forks, Fig. 3f)[5]. The lack of a functional Slx8 pathway (in *slx8-29*, *rfp1Δ*, *rfp2Δ* or double mutants) did not impede or enhance the level of resected forks (Fig. 3f, g and Supplementary Fig. 3b, c). Hence, the lack of Slx8-mediated anchorage to NPCs impedes HR-mediated DNA synthesis downstream of fork-resection and Rad51 loading, suggesting that the processing of SUMO conjugates is necessary to complete RDR.

**Nup132 promotes HR-dependent DNA synthesis in a post-anchoring manner.** To elucidate the mechanisms engaged at NPCs, we focused on the two fission yeast orthologues of Nup133, a component of the Y-shaped Nup107-Nup160 complex: Nup132 that is the most abundant (~3000 molecules/cell), and localized at the nuclear side of NPCs, whereas Nup131 is less expressed (~200 molecules/cell) and is localized at the cytoplasmic side[56]. Interestingly, *nup132Δ* cells, but not *nup131Δ* cells, were sensitive to a broad range of replication-blocking agents, including hydroxyurea (HU), but not to DSBs induced by bleomycin or to UV-induced DNA damage (Fig. 4a). A major function of NPCs being the transport of macromolecules, we further analyzed protein import and mRNA export in these mutants. Neither the absence of Nup131 nor Nup132 affected

**Fig. 3 Slx8 STUbL is necessary for anchoring to NPCs and RDR but not for safeguarding fork-integrity. a** Co-localization events in S-phase cells in indicated conditions and strains, as described on Fig. 2a. *p* value was calculated by Fisher's exact test. **b** MSD of the RFB in OFF and ON conditions in n S phase cells of *slx8-29* mutant grown at permissive (25°C, left panel) and restrictive (32 °C, right panel) temperature over indicated time interval (Δ*t*). *p* value was calculated as a one sided *t*-test based on MSD curves. Black bars correspond to SEM. **c** Diagram of constructs containing the reporter gene *ura4-sd20* (green) associated (*t-ura4sd20 < ori*) or not (*t-ura4sd20-ori*) to the RFB. The non-functional *ura4-sd20* allele, containing a 20-nt duplication flanked by micro-homology, is located downstream of the RFB. Upon activation of the RFB, a restarted fork can replicate the *ura4-sd20* and the HR-mediated non-processive DNA synthesis favors the deletion of the duplication, resulting in a functional *ura4*+ gene, generating Ura+ cells. As control, the construct devoid of RFB is used to monitor the spontaneous frequency of RS that is then subtracted to obtain the frequency of RFB-induced RS. **d** Frequency of RFB-induced Ura+ reversion in indicated strains and conditions. Each dot represents one sample from n independent biological replicate. Bars indicate mean values ± SD. *p* value was calculated by two-sided *t*-test. **e** Binding of Rad51 to the RFB in *WT* and *slx8-29* strains at indicated temperature. ChIP-qPCR results are presented as RFB ON/OFF ratio for each mutant. Distances from the RFB are presented in bp. Values are mean from three independent biological replicates ± SEM. **f** Top panel: Scheme of replication intermediates (RI) analyzed by neutral-neutral 2DGE of the *AseI* restriction fragment in RFB OFF and ON conditions. Partial restriction digestion caused by psoralen-crosslinks results in a secondary arc indicated on scheme by blue dashed lines. Bottom panels: Representative RI analysis in indicated strains and conditions. The *ura4* gene was used as probe. Numbers indicate the percentage of forks blocked by the RFB ± SD. **g** Quantification of resected forks. Values are mean of two independent biological repeats.

nuclear shape and protein import, but *nup132Δ* cells exhibited a very mild defect in mRNA export (Supplementary Fig. 4) albeit moderate when compared to the strong defect reported upon heat shock[57].

We tested the role of Nup132 in the recovery from HU-stalled forks. Strains were blocked in early S-phase by exposing exponentially growing cells to HU for 4 hours and then released into HU-free media. Flow cytometry analysis indicated that the *WT* and *nup131Δ* strains reached a G2 DNA content within 45 min after release whereas *nup132Δ* and *nup131Δ nup132Δ* cells exhibited an additional 15 min delay (Supplementary Fig. 5a, left panel). Chromosome analysis by Pulse Field Gel Electrophoresis (PFGE) showed that HU treatment prevented chromosomes from migrating into the gel because of the accumulation of replication intermediates (Supplementary Fig. 5b). Sixty minutes after release, *WT* chromosomes were able to migrate into the gel and their intensity doubled 90 minutes after release, indicating that the *WT* genome was fully duplicated and replication intermediates were resolved (Supplementary Fig. 5b, c). In contrast, chromosomes from *nup132Δ* cells showed a clear delay in their ability to migrate fully into the gel. Even 90 minutes after release, chromosomes intensity did not double, indicating that *nup132Δ* genome failed to be fully duplicated because of an accumulation of unresolved replication intermediates. Our data reveal a critical role for Nup132 in promoting DNA replication upon transient fork stalling.

We asked if Nup132 and Nup131 are involved in RDR. We detected a reduced frequency of RFB-induced RS only in the absence of Nup132 and no further reduction was observed in the double *nup131Δ nup132Δ* mutant (Fig. 4b). This defect was not correlated with a less efficient Rad51 binding to the active RFB (Fig. 4c), indicating that the early step of RDR, fork-resection and Rad51 loading, are functional. The active RFB was enriched at the NP in S-phase cells in the absence of either Nup131 or Nup132, but not in the absence of both nucleoporins (Fig. 4d and Supplementary Fig. 2). Supporting this result, the active RFB bound properly to NPCs in *nup132Δ* cells but not in the double *nup131Δ nup132Δ* mutant by ChIP (Fig. 4e). Thus, Nup132 is dispensable to anchor remodeled forks to NPCs. However, the absence of both nucleoporins may modify the NPC structure, making it inefficient for anchoring. These data reveal a novel function for NPCs in which Nup132 promotes HR-dependent DNA synthesis, downstream of Rad51 binding, in a post-anchoring manner.

HR-dependent DNA synthesis is non-processive, liable to mutation, and GCR. We monitored the rate of RFB-induced mutagenesis and GCR, including translocation and genome deletion (Supplementary Fig. 6a, b for detailed explanations)[7]. Briefly, we selected *ura4* loss events after RFB induction or not and analyzed the events by PCR to discriminate between mutation, translocation,

and genomic deletion; all these events occur in an HR-dependent manner. In *WT* cells, the induction of the RFB resulted in a 4.5, 10, and 14-fold increase in the rate of mutagenesis, deletion, and translocation, respectively (Supplementary Fig. 6c, d). The rate of translocation and genomic deletion were unaffected in the absence of Nup131 and Nup132, but RFB-induced mutagenesis was abolished in *nup131Δ* and *nup132Δ* single mutants or in the double mutant, indicating a role of both nucleoporins in promoting mutagenic HR-mediated DNA synthesis. Altogether, our data reveal a novel NPC function, via Nup132 and to a lesser extent Nup131, in promoting HR-dependent DNA synthesis. The distinct contribution of Nup131 and Nup132 to this pathway might reflect their different localization within NPCs and/or their relative abundance[56].

**Pli1-dependent SUMO chains are toxic to HR-dependent DNA synthesis.** Our data indicate that anchoring to NPCs is not sufficient to promote RDR, as exemplified in the *nup132Δ* mutant. In the absence of Nup132, the SUMO deconjugating enzyme Ulp1 is delocalized from NPCs and can no longer antagonize the PIAS family E3 ligase Pli1 that promotes 90% of bulk SUMOylation and SUMO chain formation. As a consequence, both Ulp1 and Pli1 expression are lowered, resulting in a low global SUMOylation level[35]. Surprisingly, the deletion of *pli1* partly rescued the sensitivity of *nup132Δ* cells to replication stress (Fig. 5a), suggesting a toxicity of Pli1 activity in the absence of Nup132. We asked if this toxicity might also underlie the RDR defect. The active RFB did not shift to the NP nor bound to NPCs in the absence of Pli1 (Fig. 5b, c and Supplementary Fig. 2). MSD analysis confirmed an absence of reduced mobility of the active RFB and thus a lack of anchorage in *pli1Δ* cells (Fig. 5d). However, the lack of Pli1 did not affect RFB-induced RS (Fig. 5e), indicating that RDR is fully completed without anchorage to NPCs when Pli1 is absent. Interestingly, the lack of Pli1 partly rescued the defect in RFB-induced RS of *nup132Δ* cells (Fig. 5e), even though the active RFB was still unable to bind NPCs (Fig. 5b, c). A similar rescue was observed in *slx8-29 pli1Δ* cells (Fig. 5e), consistent with Pli1 causing genome instability in the absence of STUbL activity[54,55]. Of note, the deletion of *pli1* did not rescue the mRNA export defect of *nup132Δ* cells, showing that the role of Nup132 in promoting RDR and mRNA export are uncoupled (Supplementary Fig. 4d, e). Thus, Pli1 activity is necessary to anchor arrested forks to NPCs but is toxic to HR-dependent DNA synthesis, in the absence of Nup132 and STUbL activity, suggesting a role for NPCs in counteracting this toxicity.

To gauge the type of SUMOylation involved in relocation but becoming toxic to HR-mediated DNA synthesis, we manipulated the level and type of SUMO conjugates by several means. We employed a "Low SUMO" strain in which the endogenous SUMO

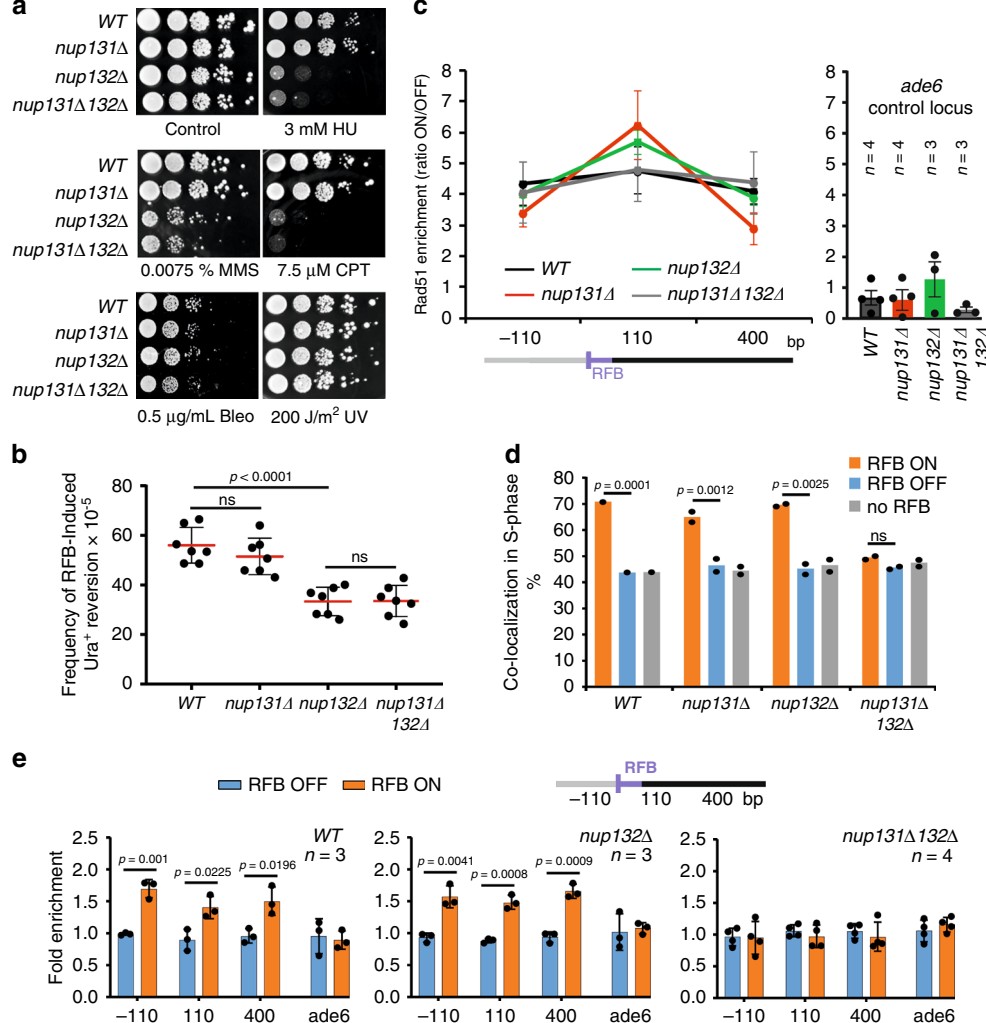

**Fig. 4 Nup132 promotes HR-mediated DNA synthesis, downstream of Rad51 binding, in a post-anchoring manner. a** Sensitivity of indicated strains to indicated genotoxic drugs. Ten-fold serial dilution of exponential cultures were dropped on appropriate plates. Bleo bleomycin; CPT camptothecin; HU hydroxyurea; MMS methyl methane sulfonate and UV: Ultra Violet-C. See supplementary Fig. 4 for the characterization of macromolecules transport and supplementary Fig. 5 for replication defect upon HU-fork stalling. **b** Frequency of RFB-induced Ura$^+$ reversion in indicated strains and conditions. Each dot represents one sample from seven independent biological replicate for each strain. Bars indicate mean values ± SD. $p$ value was calculated by two-sided $t$-test. **c** Binding of Rad51 to the RFB in indicated strains as described on Fig. 3e. Values are mean from n independent biological replicates ± SEM. **d** Co-localization event in S-phase cells in indicated conditions and strains. In all, 250 cells were analyzed for each condition and strain. $p$ value was calculated by Fisher's exact test for OFF and ON groups for each mutant and condition. Dots represent values obtained from two independent biological experiments. For each set of data, *WT* strain was analyzed alongside mutants. **e** Binding of Npp106-GFP to the RFB in indicated strains. Upstream and downstream distances from the RFB are presented in bp. Primers targeting *ade6* gene were used as unrelated control locus. Values are mean of n independent biological repeats ± SD. $p$ value was calculated using two-sided $t$-test.

promoter was replaced by a weaker constitutive promoter[53] and a *pmt3-KallR* mutant (SUMO-KallR) in which all internal Lys are mutated to Arg to prevent poly-SUMOylation[55]. Pli1-dependent SUMO chain formation is enhanced by the interaction between the single E2 SUMO conjugating enzyme Ubc9 and SUMO. Thus, we took advantage of the *pmt3-D81R* mutant (SUMO-D81R) that impairs Ubc9-SUMO interaction and allows mono and di-SUMOylation to occur in a Pli1-dependent manner but impairs the chain-propagating role of Pli1 that is toxic in the absence of STUbL[55]. In all conditions, the active RFB did not shift to the NP and RFB-induced RS was slightly increased (Fig. 5f, g), indicating that poly-SUMOylation is instrumental in relocating the RFB but impedes HR-dependent DNA synthesis. Moreover, all conditions restored RFB-induced RS in *nup132Δ* cells, indicating SUMO chains are the source of toxicity to RDR (Fig. 5g). Hence, relocation requires Pli1-dependent SUMO chain formation which

then limits HR-mediated DNA synthesis, generating a need to overcome this inhibitory effect by events occurring at NPCs. In addition, limiting the SUMO chain-propagating role of Pli1 is sufficient to bypass the necessity for relocation to NPCs to ensure efficient RDR.

**Relocation to NPCs allows SUMO chains removal by Ulp1 and the proteasome.** Relocation to NPCs is necessary to overcome the inhibitory effect of SUMO chains when priming HR-mediated DNA synthesis. STUbLs promote the ubiquitylation of SUMO conjugates for proteolysis by the proteasome, whose activity is enriched at the NP[33]. We focused on Rpn10, a regulatory subunit of the proteasome, whose absence results in defective degradation of ubiquitinated proteins[58]. In *rpn10Δ* cells, the active RFB shifted to the NP but the frequency of RFB-induced RS was severely

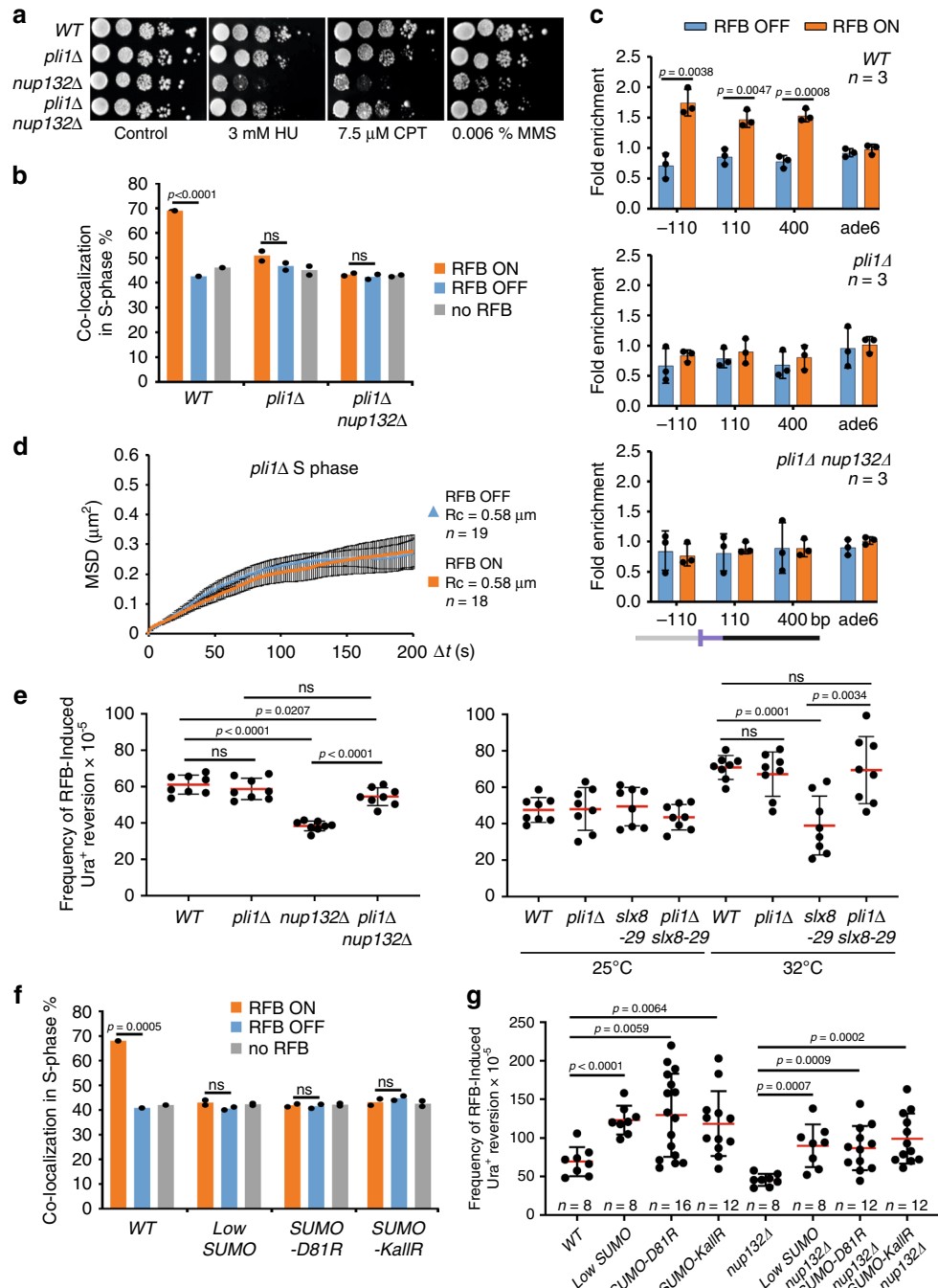

**Fig. 5 Pli1-dependent SUMO chain promotes relocation to NPCs but are toxic to RDR. a** Sensitivity of strains to indicated genotoxic drugs. Ten-fold serial dilution of exponential cultures were dropped on appropriate plates as described in Fig. 4a. **b** Co-localization event in S-phase cells in indicated conditions and strains as on Fig. 2a. $p$ value was calculated by Fisher's exact test for OFF and ON groups for each mutant and condition. **c** Binding of Npp106-GFP to the RFB in indicated strains. Upstream and downstream distances from the RFB are presented in bp. Primers targeting *ade6* gene were used as unrelated control locus. Values are mean of three independent biological repeats ± SD. $p$ value was calculated using two-sided $t$-test. **d** MSD of the RFB in OFF and ON conditions in S phase cells of *pli1Δ* mutant over indicated time interval (Δt) calculated for $n$ independent cells, as described on Fig. 1d. Black bars correspond to SEM. **e** Frequency of RFB-induced Ura+ reversion in indicated strains and conditions. Each dot represents one sample from eight independent biological replicate for each strain. Bars indicate mean values ± SD. $p$ value was calculated by two-sided $t$-test. **f** Co-localization event in S-phase cells in indicated conditions and strains as in Fig. 2a. $p$ value was calculated by Fisher's exact test for OFF and ON groups for each mutant and condition. **g** Frequency of RFB-induced Ura+ reversion in indicated strains and conditions. Each dot represents one sample from n independent biological replicate for each strain. Bars indicate mean values ± SD. $p$ value was calculated by two-sided $t$-test.

decreased and a slight additivity was observed in *nup132Δ rpn10Δ* cells (Fig. 6a, b). Thus, the proteasome activity is necessary for efficient RDR but this might not be under regulation by Nup132.

In the absence of Nup132, Ulp1 is delocalized from NPCs that are no longer able to counteract the toxicity of SUMO chains to promote RDR. Thus, we investigated the role of Ulp1 in RDR. The overexpression of Ulp1 rescued the defective RFB-induced RS of *nup132Δ* cells (Fig. 6b), indicating that low Ulp1 expression is detrimental to efficient RDR. We employed a LexA-based tethering approach to artificially target Ulp1 to the RFB[23] (Fig. 6c). Expression of Ulp1-LexA did not lead to sensitivity to genotoxic agents in striking contrast to *ulp1Δ* cells (Fig. 6d), indicating the fusion protein is functional. Ulp1-LexA was enriched in the vicinity of the RFB only in the presence of 8 LexA binding sites (at the *t-LexBS-ura4sd20 < ori* construct, Fig. 6e). Consistent with the role of Nup132 in anchoring Ulp1 at the NP, the inactive RFB shifted to the NP, in a Nup132 manner. When activated, the RFB shifted to the NP in the absence of Nup132, confirming that Ulp1 is not necessary for anchorage (Fig. 6f). Remarkably, tethering Ulp1-LexA to the active RFB, anchored to NPCs, resulted in an increased frequency of RFB-induced RS in the absence of Nup132, reinforcing the notion that Ulp1-associated NPCs are required to overcome the inhibitory effect of poly-SUMOylation on HR-mediated DNA synthesis (Fig. 6g).

**Pli1 safeguards the integrity of nascent strands at arrested forks**. A question arising from our work is the positive effect of Pli1 activity at sites of replication stress. Although *pli1Δ* cells were insensitive to replication-blocking agents, they exhibited a clear defect in the recovery from HU-stalled forks and in chromosomes duplication, suggesting an accumulation of unresolved replication intermediates (Supplementary Fig. 5). We thus investigated the integrity of the fork arrested at the RFB by 2DGE and observed an increased level of resected forks in *pli1Δ* cells (Fig. 7a, b). RPA-ChIP confirmed an extensive recruitment of RPA, up to 3 Kb upstream of the RFB, supporting the formation of larger ssDNA gaps in the absence of Pli1 (Fig. 7c). Thus, Pli1 activity is critical to negatively regulate the resection of nascent strands and safeguard fork-integrity.

## Discussion

Collapsed forks anchor to NPCs but the mechanisms engaged at NPCs to ensure fork integrity and restart were not understood. Here, we reveal the beneficial and detrimental functions of SUMOylation at replication stress sites. We propose that Pli1 activity engages at arrested forks to control the extent of nascent strand resection. Pli1 generates SUMO chains that signal for a STUbL-dependent anchorage to NPCs, but hinder the priming of HR-mediated DNA synthesis. Hence, NPCs become critical to allow the resumption of DNA synthesis by clearing off SUMO conjugates in a post-anchoring manner, via Ulp1 and proteasome activities. Selectively preventing Pli1-mediated SUMO chains bypasses the need for anchorage to NPCs while maintaining efficient RDR. Thus, SUMO-regulated mechanisms spatially segregate the subsequent steps of RDR from Rad51 loading and activity occurring in the nucleoplasm and the restart of DNA synthesis occurring after anchorage to NPCs (Fig. 7d).

We establish that DSB formation is not a requirement to anchor arrested forks to NPCs. Instead, it requires forks to be remodeled by Rad51 enzymatic activity. Relocation requires nascent strand resection to occur for Rad51 loading, but is not sufficient per se. SUMOylation of HR factors is necessary to anchor expanded CAG tracts to NPCs[59] and therefore their absence at the RFB may impair the wave of SUMOylation

necessary for relocation. However, the lack of relocation in the Rad51-II3A mutant indicates that joint-molecules, such as D-loops from which DNA synthesis is primed, are also relevant positioning signals to relocate arrested forks to NPCs. In several eukaryotes, relocation of DSBs to the NP requires end-resection and Rad51, suggesting that Rad51-mediated repair progression stabilizes repair intermediates to facilitate anchorage[59]. Breaks within repeated sequences (heterochromatin in flies, mouse peri-centromere, rDNA in budding yeast) shift away from their compartments to continue HR repair and load Rad51 at mobilized DNA damage sites[26,43,45,60]. Relocation of forks collapsed at expanded CAG repeats requires nuclease activities to engage SUMO-RPA onto ssDNA which prevents Rad51 loading. Anchorage to NPCs then facilitates Rad51 loading[59]. Here, we report a distinct situation when forks arrest within a unique sequence. Relocation requires Rad51 loading and enzymatic activity and the lack of anchorage (in STUbL or nucleoporin mutants) does not affect Rad51 loading, supporting that Rad51 loading and enzymatic activity occur prior to anchorage to NPCs. These distinct situations likely reflect different mechanisms engaged at unique sequence versus repeated sequences, where controlling Rad51 loading is of major importance to avoid potential rearrangements for the latter.

STUbL binds to SUMO modified DNA repair factors via its SIM domains to tether DNA lesions to NPCs[16,59]. Our data are consistent with this and highlight the positive and negative effects of bulk SUMOylation mediated by Pli1. Though the potential mode of Pli1 recruitment to replication stress sites remain to be identified, we show that Pli1 engagement at arrested forks is vital to safeguard fork-integrity. We noticed that the lack of Pli1 did not increase RDR efficiency whereas preventing SUMO chains does, suggesting that Pli1-dependent mono-SUMOylation events remain necessary to RDR. The Ubc9-SUMO interface may help to increase the local concentration of SUMO particles to enhance Pli1-mediated SUMO chains and mediate anchorage to NPCs. In contrast to forks collapsed at CAG tracts[59], relocation requires poly-SUMOylation as reported for persistent DSBs in budding yeast[20]. However, those SUMO chains limit HR-mediated DNA synthesis, possibly the DNA synthesis primed from D-loops, a step necessary to ensure efficient fork restart. A selective defect in Pli1-mediated SUMO chain or preventing poly-SUMOylation bypasses the need for relocation to NPCs and alleviates the toxicity of SUMO conjugates. A remaining question is whether the SUMO-targets responsible for relocation and preventing the priming of HR-mediated DNA synthesis are similar or distinct.

A possible scenario is that SUMO-dependent relocation to NPCs occurs when arrested forks are not rescued in a timely manner by opposite forks: this would lead to safeguarding fork-integrity by Pli1, and thus engaging the relocation process to NPCs. Interestingly, the lack of STUbL resulted in increased mobility of arrested forks, a phenomena not observed in the absence of Pli1, suggesting that SUMOylation promotes chromatin mobility of replication stress sites and STUbL promotes their anchorage to NPCs.

Collectively, this study uncovers how anchorage to NPCs helps to sustain DNA synthesis upon replication stress. The lack of Nup132 provides a unique genetic situation to uncouple the role of NPCs in anchoring arrested forks from their role in promoting DNA synthesis upon stress conditions. We establish that Nup132 is necessary to prime HR-mediated DNA synthesis, downstream of Rad51 binding and activity, in a post-anchoring manner. This function is linked to the role of Nup132 in recruiting Ulp1 at NPCs and is uncoupled from the transport of macromolecules. We propose that Ulp1-associated NPCs, as well as proteasome activity, are critical to remove SUMO conjugates from joint-molecules to allow DNA synthesis resumption. Consistent with

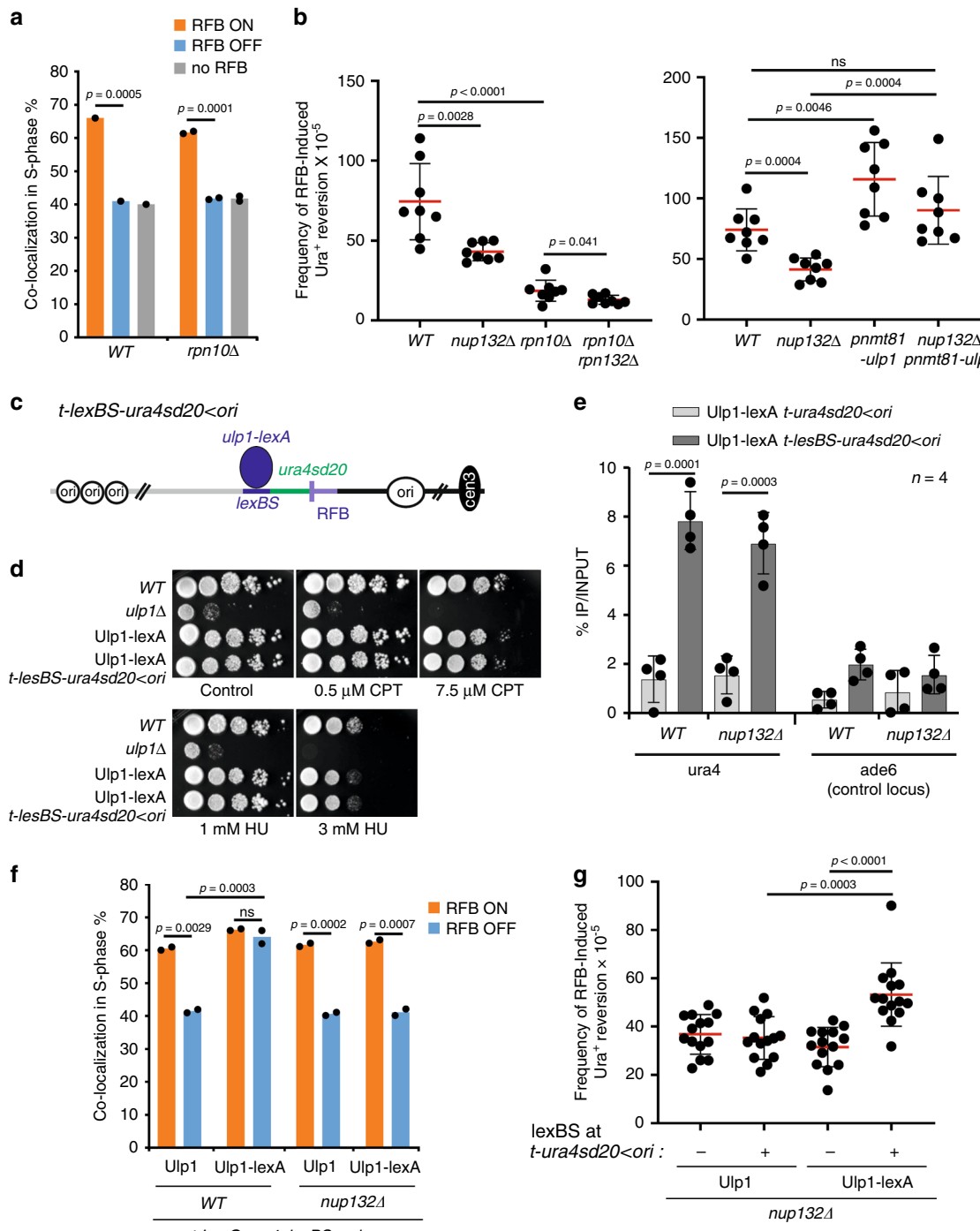

**Fig. 6 Proteasome and Ulp1 activity are necessary to clear off SUMO conjugates to promote RDR. a** Co-localization event in S-phase cells in indicated conditions and strains as on Fig. 2a. $p$ value was calculated by Fisher's exact test for OFF and ON groups for each mutant and condition. **b** Frequency of RFB-induced Ura[+] reversion in indicated strains and conditions. Each dot represents one sample from eight independent biological replicate for each strain. Bars indicate mean values ± SD. $p$ value was calculated by two-sided $t$-test. **c** Diagram of construct containing lexA-binding site (lexBS, purple) that allows tethering of Ulp1-lexA to the $t$-lexBS-ura4sd20 < ori construct (**d**, **e**, **g**) or to the $t$-Laco-ura4::lexBS < ori construct (**f**). **d** Sensitivity of indicated strains to indicated genotoxic drugs. Ten-fold serial dilution of exponential cultures were dropped on appropriate plates. **e** Binding of Ulp1-LexA to ura4 or ade6 (unrelated control locus) in the presence of LexBS ($t$-lexBS-ura4sd20 < ori) or not ($t$-ura4sd20 < ori). Values are mean of four independent biological repeats ± SD. $p$ value was calculated using two-sided $t$-test. **f** Co-localization event in S-phase cells in indicated conditions and strains as on Fig. 2a. $p$ value was calculated by Fisher's exact test for OFF and ON groups for each mutant and condition. **g** Frequency of RFB-induced Ura[+] reversion in indicated strains and conditions. Each dot represents one sample from 14 independent biological replicate for each strain. Bars indicate mean values ± SD. $p$ value was calculated by two-sided $t$-test.

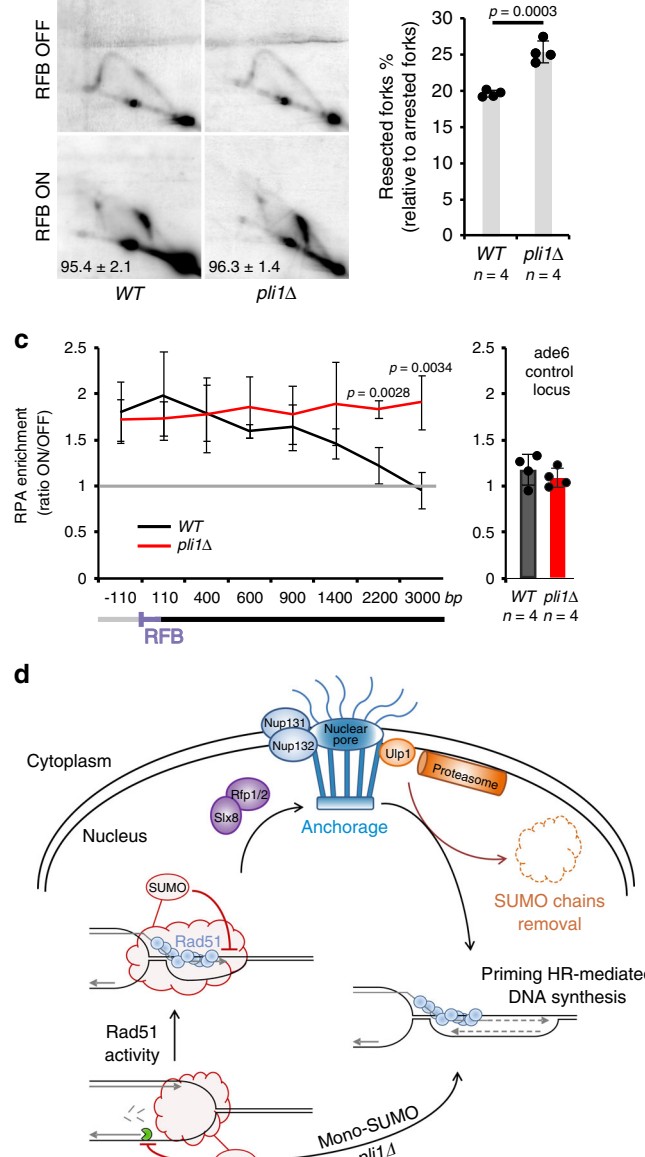

**Fig. 7 Pli1 safeguards fork-integrity by limiting resection of nascent strands. a** Representative RI analysis in indicated strains and conditions as described on Fig. 3. **b** Quantification of resected forks. Values are mean of four independent biological replicates ±SD. p value was calculated by two-sided t-test. **c** Binding of RPA (Ssb3-YFP) to the RFB in indicated strains. ChIP-qPCR results are presented as ON/OFF ratio for each mutant. Upstream and downstream distances from the RFB are presented in bp. Values are mean from four independent biological replicates ±SD. p value was calculated by two-sided t-test. Primers targeting ade6 gene were used as unrelated control locus. **d** SUMO-based regulation of relocation of remodeled forks to NPCs to promote RDR. See text for explanations.

budding yeast Nup84 sustaining fork progression at stalled forks[41], Nup132 is necessary to sustain DNA replication upon HU treatment. The deletion of Pli1 did not rescue the defect in the recovery from HU-stalled forks in nup132Δ cells (Supplementary Fig. 5), indicating that Nup132 sustains DNA replication upon stress by distinct mechanisms according to the nature of stalled versus dysfunctional forks.

SUMOylation is a dynamic and reversible modification. At dysfunctional forks, our data establish a clear role of NPCs in counteracting the toxicity of SUMO chains to allow HR-mediated

DNA synthesis. SUMO removal involves Ulp1 and the proteasome, two activities occurring at the NP. Although the role of NPCs in promoting the removal of SUMO conjugates has been previously proposed[17], our work reveals the versatile functions of SUMOylation in promoting fork integrity and relocation at the expense of limiting the step of HR-mediated DNA synthesis. We propose that SUMO-primed ubiquitylation promotes the clearance of DNA repair/replication factors at arrested forks to prime DNA synthesis, but the multiple targets remain unknown. Interestingly, the Branzei lab recently identified replication factors undergoing SUMOylation regulated by Ulp2 and STUbL to control replication initiation[61]. Similarly, we propose that key SUMOylated factors are controlled by Ulp1 and STUbL to regulate timely fork restart.

## Methods

**Standard yeast genetics**. Yeast strains and primers used in this work are listed in Table S1 and S2 respectively. Gene deletion or tagging were performed by classical genetic techniques. Strain with SUMO-KallR was obtained by integration of synthetized mutated pmt3 gene (Genscript) into pmt3::ura4 and colonies were selected on 5-FOA. Mutation of all lysines to arginines was confirmed by sequencing. To assess the sensitivity of chosen mutants to genotoxic agents, midlog-phase cells were serially diluted and spotted onto plates containing hydroxyurea (HU), methyl methanesulfonate (MMS), campthotecin (CPT), bleomycin (bleo) or irradiated with an appropriate dose of UV. Strains carrying the RTS1, replication fork block sequence were grown in minimal medium EMMg (with glutamate as nitrogen source) with addition of appropriate supplements and 60 μM thiamine (barrier inactive, OFF). The induction of replication fork block was obtained by washing away the thiamine and further incubation in fresh medium for 24 h (barrier active, ON).

**Live cell imaging**. For snapshot microscopy, cells were grown in filtered EMMg with or without 60 μM thiamine for 24 h to exponential phase (RFB OFF and RFB ON), then centrifuged and resuspended in 500 μL of fresh EMMg. In all, 1 μL from resulting solution was dropped onto Thermo Scientific slide (ER-201B-CE24) covered with a thin layer of 1.4% agarose in filtered EMMg. 21 z-stack pictures (each z step of 200 nm) were captured using 3D LEICA DMRXA microscope, supplied with CoolSNAP monochromic camera (Roper Scientific) under 100X oil-immersion magnification with numerical aperture 1.4. Exposure time for GFP channel was 500 ms, for mCherry 1000 ms. Pictures were collected with META-MORPH software and analyzed with ImageJ software. Foci that merged or partially overlap were counted as colocalization event.

The mobility of arrested forks was investigated by collecting 3-dimensional 14-stack images every 1.5 s over 5 min. Cells were visualized with a Spinning Disk Nikon inverted microscope equipped with the Perfect Focus System, Yokogawa CSUX1 confocal unit, Photometrics Evolve512 EM-CCD camera, 100X/1.45-NA PlanApo oil immersion objective and a laser bench (Errol) with 491 diode laser, 100 mX (Cobolt). Images were captured every 1.5 s with 14 optical slices (each z step of 300 nm), 100 ms exposure time for single GFP channel at 15% of laser power using METAMORPH software. Time-lapse movies were mounted and analyzed with ImageJ software as described below.

To study the colocalization time between lacO/LacI RFB foci and Npp106-GFP cells grown in the above conditions were visualized with a Nikon inverted microscope described above, using two fluorescent channels with 491 and 561 nm diode lasers, 100 mX (Cobolt). Images were captured every 10 s with 14 optical slices (each z step of 300 nm) for 30 min with 100 ms exposure time both for GFP and mCherry channels at 15% of laser power using METAMORPH software. Time-lapse movies were mounted and analyzed with ImageJ software (description below).

Protein import-export from nucleus was monitored using WT and nup131Δnup132Δ strains expressing genomic LacI-NLS-GFP without LacO repeats integrated into the genome. Cells grown for 24 h with or without thiamine were visualized with Nikon inverted microscope described above. Snapshot pictures (21 stacks, each z of 200 nm and 100 ms exposure) were acquired using METAMORPH software and analyzed in ImageJ. Images were projected for maximum intensity. The nuclear/cytoplasmic ratio (N/C) was determined by measuring mean fluorescence intensity within constant square regions (ROI plugin from ImageJ) placed in the cytoplasm, center of nucleus and intercellular background. Nuclear/cytoplasm ratio stand for (Nucleus-background)/(Cytoplasm-background).

All image acquisition was performed on the PICT-IBiSA Orsay Imaging facility of Institut Curie.

**Movie analysis**. Movies have been mounted using ImageJ. For analysis of mobility of arrested forks after projection around z-axis, single-particle tracking was performed using ImageJ plugin SpotTracker[62]. Obtained coordinates for RFB foci were

then analyzed using MS Excel macro as in ref. [63]. Derived parameters were utilized to calculate mean square displacement and radius of constrains for each condition. The statistical significance was calculated based on MSD curves by one-tailed $t$-test.

For co-localization analysis of RFB foci and Npp106-GFP, first z projection was done for GFP and mCherry channels, then pictures have been denoised by subtracting background, compensated for bleaching over time (ImageJ plugin Stack Contrast Adjustment) and finally processed with filter Gaussian Blur. A kymograph was constructed over each S phase nucleus in indicated strains (Supplementary Fig. 1). First, all optical slices were projected around z-axis using average intensity parameter. Then a resulting 2D movie, consisting of 181 frames of 10 s interval was analyzed to pick up cells forming septum, which undergo S phase for analysis (Supplementary Fig. 1). Subsequently a projection over time-axis to form kymograph for GFP and mCherry channels was made. Next, to investigate the colocalization between Npp106-GFP-stained nuclear periphery and lacO-bound mCherry-LacI both channels were merged. The time of colocalization has been estimated based on the overlap of RTS1-RFB lacO/lacI-mCherry signal and Npp106-GFP signal (Supplementary Fig. 1).

**Chromatin immunoprecipitation.** ChIP against Npp106-GFP, Man1-GFP, Sad1-GFP, as well as RPA (ssb3-YFP) were performed as described in ref. [4] with following modifications. 200 mL of culture (at $1 \times 10^7$ concentration) for each condition (RTS1-RFB OFF/ON) was divided into $2 \times 100$ mL aliquots and then crosslinked with 10 mM DMA (dimethyl adipimidate, thermo scientific, 20660) and subsequently 1% formaldehyde (Sigma, F-8775). Next, cells from each 100 mL were then frozen in liquid nitrogen and lysed by bead beating in 400 µL of lysis buffer (50 mM HEPES pH 7.5, 1% Triton X100, 0.1% Nadeoxycholate, 1 mM EDTA with 1 mM PMSF and Complete EDTA-free protease inhibitor cocktail tablets (Roche, 1873580). Chromatin sonication was performed using a Diagenode Bioruptor in a mode High, 10 cycles of 30 s ON and 30 s OFF. Then sonicated chromatin fractions were pooled (400 µL + 400 µL) for each condition and immunoprecipitation over night was performed as follows: 300 µL was incubated with anti-GFP antibody (Invitrogen, A11122) at 1:150 concentration, 300 µL was incubated with Normal Rabbit IgG antibody (Cell Signaling Technology, #2729S) at concentration 1:75 and 5 µL was preserved as INPUT fraction. Next morning a Protein G Dynabeads (Invitrogen, 10003D) were added for 1 h and immunoprecipitated complexes have been decrosslinked for 2 h at 65° C. The DNA associated with respective protein was purified with a Qiaquick PCR purification kit (QIAGEN, 28104) and eluted in 400 µL of water. qPCR (iQ SYBR green supermix, Biorad, 1708882, primers listed in Table S2) was performed to determine the relative amounts of DNA (starting quantities based on standard curves for each pair of primers) using BIORAD CFX Maestro v1.1. For Npp106-GFP, Man1-GFP and Sad1-GFP enrichment, based on starting quantities, was normalized by dividing specific GFP signal over rabbit IgG control and then relative to an internal control locus at chromosome II (II.50). RPA enrichment, based on starting quantities, was calculated in the same way as Npp106-GFP, Man1-GFP, Sad1-GFP and presented as ratio ON/OFF.

Rad51 and Ulp1-lexA ChIP were performed with above protocol, but rad51Δ or strain devoid of Ulp1-lexA were used as a negative control, instead of normal rabbit IgG antibody[6]. Briefly, cells cross-linked with DMA and 1 % formaldehyde were subjected to ChIP protocol and immunoprecipitation was performed overnight using anti-Rad51 antibody (Abcam, ab63799) at 1:300 dilution or anti-lexA antibody (Abcam, ab174384) at 1:120 dilution. For qPCR starting quantities have been determined and the enrichment was calculated by subtracting negative control values and internal control locus at chromosome II (II.50).

**2DGE analysis of replication intermediates.** Exponential cells ($2.5 \times 10^9$) were treated with 0.1% sodium azide and subsequently mixed with frozen EDTA (of final concentration at 80 mM). Genomic DNA was crosslinked with trimethyl psoralen (0.01 mg/mL, TMP, Sigma, T6137) added to cell suspensions for 5 min in the dark. Next, cells were irradiated with UV-A (365 nm) for 90 s at a constant flow 50 mM/cm$^2$. Subsequently, cell lysis was performed by adding lysing enzymes (Sigma, L1412) at concentration 0.625 mg/mL and zymolyase 100 T (Amsbio, 120493-1) at 0.5 mg/mL. Obtained spheroplasts were next embedded into 1 % low melting agarose (InCert Agarose 50123, Lonza) plugs and incubated overnight at 55 °C in a digestion buffer with 1 mg/mL of proteinase K (Euromedex EU0090). Then plugs were washed with TE buffer (50 mM Tris, 10 mM EDTA) and stored at 4 °C. Digestion of DNA was performed using 60 units per plug of restriction enzyme AseI (NEB, R0526M), next samples were treated with RNase (Roche, 11119915001) and beta-agarase (NEB, M0392L). Melted plugs were equilibrated to 0.3 M NaCl concentration. Replication intermediates were purified using BND cellulose (Sigma, B6385) poured into columns (Biorad, 731-1550)[10]. RIs were enriched in the presence of 1 M NaCl 1.8% caffeine (Sigma, C-8960), precipitated with glycogen (Roche, 1090193001) and migrated in 0.35% agarose gel (1xTBE) for the first dimension. The second dimension was cast in 0.9% agarose gel (1xTBE) supplemented with EtBr. Next DNA was transferred to a nylon membrane (Perkin-Elmer, NEF988001PK) in 10x SSC. Finally, membranes were incubated with $^{32}$P-radiolabeled ura4 probe (TaKaRa BcaBEST$^{TM}$ Labeling Kit, #6046 and alpha-$^{32}$P dCTP, Perkin-Elmer, BLU013Z250UC) in Ultra-Hyb buffer (Invitrogen, AM8669) at 42 °C. Signal of replication intermediates was collected in phosphor-imager software (Typhoon-trio) and quantified by densitometric analysis with

ImageQuantTL software (GE healthcare). The 'tail signal' was normalized to the overall signal corresponding to arrested forks.

**Replication slippage assay.** The frequency of ura4+ revertants using ura4-sd20 allele was performed as follows. 5-FOA (EUROMEDEX, 1555) resistant colonies were grown on plates containing uracil with or without thiamine for 2 days at 30 °C and subsequently inoculated into EMMg supplemented with uracil for 24 h. Then cultures were diluted and plated on EMMg complete (for cell survival) and on EMMg without uracil but supplemented with 60 µM thiamine. After 5–7 days incubation at 30 °C colonies were counted to determine the frequency of ura4+ reversion. To obtain the true occurrence of replication slippage by the RTS1-RFB, independently of the genetic background, we subtracted the replication slippage frequency of the strain devoid of RFB (considered as spontaneous frequency) from the frequency of the strain containing the t-ura4sd20 < ori construct, upon expression of Rtf1.

**Flow cytometry.** Flow cytometry analysis of DNA content was performed as follows[64]: cells were fixed in 70% ethanol and washed with 50 mM sodium citrate, digested with RNAse A (Sigma, R5503) for 2 h, stained with 1 µM Sytox Green nucleic acid stain (Invitrogen, S7020) and subjected to flow cytometry using FACSCANTO II (BD Biosciences). Gating procedure is presented on Supplementary Fig. 5.

**Pulse field gel electrophoresis.** Yeast cultures were grown to logarithmic phase in rich YES medium to concentration $5 \times 10^6$/mL, synchronized in 20 mM HU for 4 hours, subsequently released to fresh YES medium. At each time point 20 mL of cell culture was harvested, washed with cold 50 mM EDTA pH 8 and digested with lyticase (Sigma, L4025) in CSE buffer (20 mM citrate/phosphate 5.6, 1.2 M sorbitol, 40 mM EDTA pH 8). Next cells were embedded into 1% UltraPure$^{TM}$ Agarose (Invitrogen, 16500) and distributed into 5 identical agarose plugs for each time point. Plugs were then digested with Lysis Buffer 1, LB1 (50 mM Tris-HCl pH 7.5, 250 mM EDTA pH 8, 1 % SDS) for 1.5 hour in 55 °C and then transferred to Lysis Buffer 2, LB2 (1% N-lauryl sarcosine, 0.5 M EDTA pH 9.5, 0.5 mg/mL proteinase K) o/n at 55 °C. Next day LB2 was change for fresh one and digestion was continued o/n at 55 °C. After, plugs were kept at 4 °C. To visualize intact chromosomes one set of plugs was run on a Biorad CHEF-DR-III pulse field gel electrophoresis (PFGE) system for 60 h at 2.0 V/cm, angle 120°, 14 °C, 1800 s single switch time, pump speed 70 in 1x TAE buffer. Separated chromosomes were stained in ethidium bromide (10 µg/mL) for 30 min, washed briefly in 1x TAE and visualized with UV transilluminator.

**Ura4 loss assay.** Mutant strains were grown on complete EMMg plates with or without 60 µM thiamine. Then 11 independent colonies from each strain and condition were inoculated into 5 mL of complete EMMg with or without thiamine and grown to stationary phase. Appropriate dilutions were plated on YES plates (for cell survival) and on 0.1% 5-FOA (EUROMEDEX, 1555) plates. After 5–7 days incubation at 30 °C colonies were counted. The rate of ura4+ loss was determined by the median and statistical significance was measured by nonparametric Mann–Whitney U test.

**PCR assays for determination of the rates of genomic deletion, translocation, and mutation.** 100 5-FOA resistant colonies per strain per condition were subjected for PCR analysis (primers for translocation junction, ura4+ and control gene rng3+ listed in Table S2) as reported in ref. [7]. Translocations, deletions and mutations were counted as percentages of all events and these values were used to balance the rates of ura4 loss and subsequently to estimate the respective rates of translocations, deletions and mutations. Mann–Whitney U test was used to check the statistical significance of analyzed data.

**Fluorescence in situ hybridization.** Fluorescence in situ hybridization (FISH) was performed as described elsewhere[65] with following modifications. Strains were grown in complete EMMg media without thiamine and fixed with formaldehyde (Sigma, F8775) added to the final concentration of 4% for 45 min with rotation at RT. Next, cells were washed twice in Fixation buffer (1.2 M sorbitol, 100 mM KHPO$_4$, pH 7.5) and resuspended in fresh Fixation buffer containing 100 T zymolyase (MP Biomedicals, SKU08320932) at final concentration 0.5 µg/mL. Cells were then digested for 10 min at 30 °C. Spheroplasts were gently washed twice with ice-cold Fixation buffer. Next cells were stored in 70% ethanol for at least 3 h at 4 °C. Then ethanol was carefully removed after mild centrifugation (400 g) and cells were incubated for 5 min, RT in 2x SSC. Subsequently, cells were resuspended in Hybridization buffer (50% formamide, 10% dextran sulphate, 4x SSC, 0.02% polyvinyl pyrrolidone, 0.02% BSA, 0.02% Ficoll 400, 125 µg/mL of E. coli tRNA, 500 µg/mL salmon sperm DNA) and prehybridized at 37 °C for 1 h. Hybridization overnight, in the dark, was performed in Hybridization buffer with 10 µg/mL of Oligo-dT$_{50}$-Cy3 probe at 37 °C. Next day cells were incubated at RT for 30 min in 2X SSC and then 30 min 1X SSC. The last incubation with 0.5X SSC was carried at 37 °C. Subsequently cells were incubated with DAPI solution (diluted 1/4000 in 0.5x SSC) for 3 min. Next cells were washed with 0.5X SSC for 5 min to remove

excess of DAPI. Finally, cells were resuspended in 15 μL of VECTASHIELD® Antifade Mounting Medium and subjected to snapshot microscopy on glass slides using a DM6000B Leica microscope with a 100×, NA 1.4 (HCX Plan-Apo) oil immersion objective coupled to a piezo-electric motor (LVDT; Physik Instrument) and a CCD camera (CoolSNAP HQ; Photometrics). In all, 21 z-stacks of 200 nm, with 300 ms exposure time for Cy3 and 50 ms for DAPI channels were collected with METAMORPH and analyzed with ImageJ software. Percentage of cells with poly(A) + RNA accumulation was calculated from at least 200 cells per each strain and condition.

**Statistical analysis**. Quantitative densitometric analysis of Southern-blots after 2DGE was carried using ImageQuant software. The 'tail signal' of resected forks was normalized to the overall signal corresponding to arrested forks.

Quantification of PFGE was performed using ImageJ and presented as percenatge of migrating chromosomes relative to asynchronous profile.

Cell imaging was performed using METAMORPH software and processed and analyzed using ImageJ software.

The explanation and definitions of values and error bars are mentioned within the figure legends. Most experiments the number of samples is $n > 3$ obtained from independent experiments to ensure biological reproducibility. For all experiments based on the analysis of cell imaging, the number of nuclei analyzed is mentioned in the figure legends.

Statistical analysis was carried using Mann–Whitney U tests and Student's t-test.

**Reporting summary**. Further information on research design is available in the Nature Research Reporting Summary linked to this article.

## Data availability

Data have been deposited to Mendeley data and are available at: https://doi.org/10.17632/4m7z3gy5yc.1. All relevant data are available from the corresponding author. Source data are provided with this paper.

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

## Acknowledgements

We thank Joe Murray, Antony Carr, Dorota Dziadkowiec and Vincent Dion for exchanging reagents, and Xiaofeng Allen Su for performing initial pilot zoning assays with the RFB system. The Sad1-GFP strain was obtained from YGRC/NBRP Japan resource database (http://yeast.nig.ac.jp/yeast/top.xhtml). We are very grateful to Michael Boddy for providing *slx8-29*, "Low Sumo" and "SUMO-D81R" strains. We also thank the PICT-IBiSA@Orsay Imaging Facility of the Institut Curie and the Flow Cytometry Facility of the Orsay site of Institut Curie. We thank Marie-Noelle Simon and Vincent Géli for their criticisms and helpful comments on this work. This study was supported by grants from the Institut Curie, the CNRS, the *Fondation ARC* (Projet Fondation ARC PJA 20181208114), the *Fondation pour la Recherche Médicale* "Equipe FRM DEQ20160334889", LIGUE contre le cancer "Equipe Labellisée 2020 (EL2020LNCC/Sal), and the ANR grant NIRO (ANR-19-CE12-0023-01). AAS was funded by a French governmental fellowship and a 4th-year PhD grant from *Fondation ARC*, KK received a Postdoctoral fellow ship from *Fondation ARC* (PDF20171206749) and K.S. received a PhD fellowship from the LIGUE contre le cancer. C.H.F. is funded by NIH grant GM122880. The funders had no role in study design, data collection and analysis, the decision to publish, or preparation of the paper.

## Author contributions

K.K., K.S., V.B., and A.A.S performed the experiments. B.P., K.K., C.F., and S.A.E.L contributed to experimental design and data analysis. C.L. provided expertise to perform MSD. K.K., and S.A.E.L wrote the paper. S.A.E.L and C.H.F edited the paper.

## Competing interests

The authors declare no competing interests.
