## [Peer Review File · Nature Communications]

REVIEWER COMMENTS

Reviewer #2 (Remarks to the Author):

Kramarz et al. investigate the relationship between dysfunctional replication forks and the nuclear periphery. The authors make use of an RTS1-inducible replication fork barrier (RFB) to block a single replisome on fission yeast chromosome 3. This studied model locus exhibits increased association with the nuclear pore complex upon RFB induction. The authors identified a multi-step process in which dysfunctional replication forks are first processed, then relocated to nuclear periphery to restart DNA synthesis. The initial steps for relocation depend on short range DNA resection and homologous recombination factors, SUMO chain formation by the Pli1 ligase and the STUbL complex. At the nuclear pore complex, the SUMO-protease Upl1 and the proteasome enable the removal of SUMO chains to promote fork-restart. The authors also identified that preventing SUMO chain formation bypasses the requirement of relocation to the nuclear periphery to promote fork restart, at the expense of replication fork integrity.

Overall, Kramarz et al. identify and report novel findings with potential impact on genome stability. Particularly, the study provides unique insights into the processes mediating dysfunctional fork relocation and restart at the nuclear periphery. However, the current version of the manuscript was challenging to read, requires significant streamlining of the text, and there are important points that need to be fully addressed before the conclusions can be deemed to be fully supportive of the claims and conclusions.

Major points:

1) Writing: The abstract does not do the findings justice and I would highly recommend that the authors focus on the abstract's accessibility while better highlighting the novelty of the findings. The rest of the text could also greatly benefit from careful streamlining.

2) Some of the results and transitions presented in Figure 2 are confusing. First, the authors claim that short range resection (mediated by Ctp1 and Rad50) is required to promote relocation of the active RFB to the nuclear periphery. However, this conclusion is not supported by subsequent analyses of the double mutants *ctp1Δ pku70Δ* and *rad50Δ pku70Δ*. As mentioned by the authors, "the lack of Pku70 bypasses the requirement of MRN-Ctp1 to initiate fork resection, and that the level resected forks was fully restored in *ctp1Δ pku70Δ* and *rad50Δ pku70Δ* cells" [Teixeira-Silva et al, Nat Commun, (2017)]. Based on this observation, one alternative conclusion is that Ctp1 and Rad50 promote the relocation of the active RFB to nuclear periphery independently of resection activity. Subsequently, the authors then suggest that the failure of these double deletions (*ctp1Δ pku70Δ* and *rad50Δ pku70Δ*) to relocate active RFB to the nuclear periphery is caused by defects in replicative/recombination intermediates formed during RDR. However, this claim is not directly supported by the data. Moreover, the recruitment of Rad51 to the active RFB was shown to be significantly reduced in *pku70Δ* cells [Teixeira-Silva et al, Nat Commun, (2017)]. Yet, the active RFB in *pku70Δ* cells does not fail to relocate to the nuclear periphery, further challenging the current interpretation. Overall, I recommend that the authors better explain the logical transitions related to the results in Figure 2 while the addressing these specific points.

3) The authors claim that the anchorage of the active RFB to the nuclear pore complex is not sufficient to promote RDR. This claim is based on the significant decrease of RFB-induced replication slippage in *nup132Δ* cells (Figure 4e), even if the active RFB still localizes to the nuclear periphery in *nup132Δ* cells (Figure 4b). One alternative interpretation for the defect in recombination-dependent DNA synthesis observed in *nup132Δ* cells is that anchorage of the active RFB at the nuclear pore complex

may not be fully maintained in a *nup132Δ* background. The authors should provide a more direct approach (in addition to the microscopy-based co-localization experiments shown in figure 4b) to confirm the stable association/anchorage of the active RFB at the nuclear pore complex in a *nup132Δ* background.

4) Disruption of both Nup131 and Nup132 are required to prevent relocation of the active RFB to the nuclear periphery. Based on this result, the authors “concluded that Nup131 and Nup132 redundantly act as anchorage sites at NPC”. However, as previously shown [Asakawa et al. PLoS Genet. (2019)] and mentioned in the manuscript, Nup131 is located on the cytoplasmic side of the nuclear pore complex, raising doubt about the conclusion that Nup131 provides an anchorage site for the active RFB.

5) Preventing SUMO chain formation (Low SUMO and SUMO-DR81) and overexpressing the SUMO protease Ulp1 resulted in an increased frequency of RFB-induced Ura⁺ reversion as compared to WT (Figures 5f and 6d), whereas *pli1Δ* did not (Figure 5d). This result suggests that low levels of SUMO are in fact beneficial (as compared to no SUMO) in terms of promoting fork restart without relocation to the nuclear periphery. Please address/explain.

Minor issues:

6) For the replication slippage assay, what is the frequency of spontaneous replication slippage (i.e. without RFB, as shown in figure 2c, top panel) for all the genetic backgrounds in which this assay was performed. Also, how were the data normalized relative to WT in order to account for a possible difference in the frequency of spontaneous replication slippage among the different mutant backgrounds.

7) For Figure 5a, deletion of *pli1* only partly rescued the phenotype. This should be more clearly stated.

8) For Figure 5d, the authors stated that the “lack of Pli1 fully rescued the defect in FRB-induced RS in *nup132Δ* cells”. Statistical analysis should be performed to compare WT and *pli1Δ nup132Δ* cells.

Reviewer #3 (Remarks to the Author):

“The nuclear pore primes recombination-dependent DNA synthesis at arrested forks by promoting SUMO removal” by Kramarz et al.

In this paper by the lab of Sarah Lambert the authors use their well-described system of an inducible replication fork barrier in fission yeast to investigate the translocation/retention of stalled replication forks to the nuclear pore complex (NPC). For DSBs an important role of NPCs and other anchors to the NE has been discovered initially in budding yeast by pioneering studies of the Gasser and Jentsch lab and a clear link to the cellular SUMO machinery has been established, but some aspects have remained phenomenological. A key strength of the present study is that it extends these findings to stalled replication forks and that it genetically characterizes at so far unprecedented level the positive and negative role of SUMOylation in the process.

The data are indeed of high technical quality and the interpretations logical. The writing is in parts very condensed and the authors may want to choose which data to concentrate on in their presentation to make this paper more accessible for non-expert readers. Overall, I think this study can be suitable for publication after the authors experimentally address the following points.

(1) Presentation: a more detailed description of the experimental strategy is needed in places. For

example: the experimental approach to visualize S-phase specific relocation of the RFB site is described at length, while the genetic replication restart assay is mentioned only very briefly.

(2) The *fml1* data appears to be only very loosely connected to the rest of the paper and disrupts the logical flow. Unless the authors can provide a good explanation why it should be included, I would consider removing it from the current paper.

(3) I take issue with some of the ChIP data in this paper as the authors test exclusively genomic locations that are in immediate vicinity. Can they include additional qPCR data measuring enrichment outside the affected area, to show that the enrichment they observe is specific for the surrounding RFB location.

(4) Overall, I think the Npp106 ChIP data is an essential addition to the fluorescence microscopy data. It needs to be added for several key strains such as *nup132* and the *nup132 pli1* double mutant.

(5) In their study the authors have reached a good understanding of the *nup132* mutant phenotype. However, the phenotype of the *nup132 nup131* double mutant is much less understood. Can the authors provide further insights on this mutant? Can they really interpret its phenotype in the context of a recruitment defect or is it possible (likely?) that this mutant leads to a defect in NPC assembly and leads to for example NPC clustering?

(6) I am not convinced the authors can conclude that it is specifically SUMO chains that cause toxicity in the absence of Nup132. In order to make this conclusion a SUMO KR all (non-SUMOylatable) mutant would be required.

(7) The HU/MMS sensitivity of *nup132* mutants cannot be rescued by the *pnmt81-ulp1* strains. Is this the *pnmt81-ulp1* condition well understood? Perhaps the authors should concentrate fully on the *lexA-Ulp1* strain in the context of their RFB system, but present a full analysis here including ChIP data.

We would like to thank the two reviewers for their enthusiasm and valuable comments that have helped us to clarify our findings and strengthen our conclusion. Before addressing their specific comments point-by-point, we summarize here for the two reviewers the points that we have experimentally addressed and the fact that we have extensively revised both the textual part and some of the main figures, while respecting the space limits.

- We performed CHIP-qPCR to address the binding of NPCs to the RFB in the most relevant genetic backgrounds and to address the tethering of Ulp1-LexA to the RFB locus. These experiments reinforce our microscopy approaches.
- We tested a SUMO mutated form in which all Lysines are mutated to Arginine (KallR) to clearly establish that polySUMOylation is necessary for relocation but impedes the step of HR-mediated DNA synthesis.

Reviewer #2

1) Writing: The abstract does not do the findings justice and I would highly recommend that the authors focus on the abstract's accessibility while better highlighting the novelty of the findings. The rest of the text could also greatly benefit from careful streamlining.

The abstract is now deeply revised, as well as the main text. We hope that these changes improve the clarity and the rationale of the paper.

2) Some of the results and transitions presented in Figure 2 are confusing. First, the authors claim that short range resection (mediated by Ctp1 and Rad50) is required to promote relocation of the active RFB to the nuclear periphery. However, this conclusion is not supported by subsequent analyses of the double mutants *ctp1Δ pku70Δ* and *rad50Δ pku70Δ*. As mentioned by the authors, "the lack of Pku70 bypasses the requirement of MRN-Ctp1 to initiate fork resection, and that the level resected forks was fully restored in *ctp1Δ pku70Δ* and *rad50Δ pku70Δ* cells" [Teixeira-Silva et al, Nat Commun, (2017)]. Based on this observation, one alternative conclusion is that Ctp1 and Rad50 promote the relocation of the active RFB to nuclear periphery independently of resection activity. Subsequently, the authors then suggest that the failure of these double deletions (*ctp1Δ pku70Δ* and *rad50Δ pku70Δ*) to relocate active RFB to the nuclear periphery is caused by defects in replicative/recombination intermediates formed during RDR. However, this claim is not directly supported by the data. Moreover, the recruitment of Rad51 to the active RFB was shown to be significantly reduced in *pku70Δ* cells [Teixeira-Silva et al, Nat Commun, (2017)]. Yet, the active RFB in *pku70Δ* cells does not fail to relocate to the nuclear periphery, further challenging the current interpretation. Overall, I recommend that the authors better explain the logical transitions related to the results in Figure 2 while addressing these specific points.

We are grateful to the reviewer for this comment. First, we addressed the binding of Npp106-GFP to the RFB by CHIP-qPCR in a strain whose *pku70* ORF was replaced by an antibiotic marker (Teixeira-Silva Nature communications 2017). We observed no binding of the active RFB to NPCs. This result contrasted with our microscopy data showing a clear shift of the active RFB to the nuclear periphery. This last experiment was done in a strain whose *pku70* gene was interrupted by the *leu2⁺* marker (Tomita et al. MCB 2003). We have therefore repeated this microscopy experiment in a strain whose *pku70* ORF was replaced by an antibiotic marker and we observed no shift of the active RFB to the nuclear periphery. We suspect that a truncated form of Pku70 remains expressed in the interrupted strain and would be sufficient to trigger relocation to the nuclear periphery. Although these results open new perspectives to understand how KU mediates relocation, they clearly contradict our initial observation and conclusion. We sincerely apologize for this.

In the revised version, we provide novel microscopy data and ChIP-qPCR to establish that relocation of the active RFB to NP and binding to NPCs require Pku70 and Rad50 but not Exo1 (Fig 2a and b). We concluded that there is a lack of correlation between the extent of fork-resection and the capacity of the active RFB to bind to NPCs (page 7 lines 212-214). We also provide data to establish that the active RFB is not bound by NPCs in the *rad51-ll3A* mutant (Fig 2b and c), reinforcing our conclusion that the formation of joint-molecules at arrested forks is necessary for a stable association with NPCs (page 8 lines 225-226). Since, MRN-Ctp1 is active in the Rad51-ll3A strain, we do not favor a role of this complex in promoting relocation independently of resection. Instead, we propose that relocation requires nascent strand resection to occur for Rad51 loading, but is not sufficient *per se* (page 8 lines 224-225; and discussion page 13 lines 409-410). We discuss our finding in the context of the recent paper published by Whalen et al. (cell report 2020) in which the authors propose that relocation of forks collapsed at expanded CAG repeats requires nuclease activities to engage SUMO-RPA onto ssDNA which prevents Rad51 loading (page 13 lines 418-420). We hope that these novel data are convincing enough.

3) The authors claim that the anchorage of the active RFB to the nuclear pore complex is not sufficient to promote RDR. This claim is based on the significant decrease of RFB-induced replication slippage in *nup132Δ* cells (Figure 4e), even if the active RFB still localizes to the nuclear periphery in *nup132Δ* cells (Figure 4b). One alternative interpretation for the defect in recombination-dependent DNA synthesis observed in *nup132Δ* cells is that anchorage of the active RFB at the nuclear pore complex may not be fully maintained in a *nup132Δ* background. The authors should provide a more direct approach (in addition to the microscopy-based co-localization experiments shown in figure 4b) to confirm the stable association/anchorage of the active RFB at the nuclear pore complex in a *nup132Δ* background.

We performed ChIP-qPCR against Npp106-GFP and found that the active RFB is stably anchored to NPCs in the absence of Nup132 but not in the absence of both Nup131 and Nup132 (Fig 4e). These data reinforce our conclusion about a role of Nup132 in promoting HR-mediated DNA synthesis in a post-anchoring manner and that anchorage is not sufficient to promote RDR in the absence of Nup132. We hope that these novel data will convince the reviewer.

4) Disruption of both Nup131 and Nup132 are required to prevent relocation of the active RFB to the nuclear periphery. Based on this result, the authors “concluded that Nup131 and Nup132 redundantly act as anchorage sites at NPC”. However, as previously shown [Asakawa et al. PLoS Genet. (2019)] and mentioned in the manuscript, Nup131 is located on the cytoplasmic side of the nuclear pore complex, raising doubt about the conclusion that Nup131 provides an anchorage site for the active RFB.

We agree with this comment and we toned our conclusion down. We added the following sentence: “Thus, Nup132 is dispensable to anchor remodeled forks to NPCs. However, the absence of both nucleoporins may modify the NPC structure, making it inefficient for anchoring” (page 9 lines 287-289).

5) Preventing SUMO chain formation (Low SUMO and SUMO-DR81) and overexpressing the SUMO protease Ulp1 resulted in an increased frequency of RFB-induced Ura⁺ reversion as compared to WT (Figures 5f and 6d), whereas *pli1Δ* did not (Figure 5d). This result suggests that low levels of SUMO are in fact beneficial (as compared to no SUMO) in terms of promoting fork restart without relocation to the nuclear periphery. Please address/explain.

The absence of Pli1 cannot be considered as “no SUMO” since we cannot exclude an involvement of the second E3 SUMO ligase Nse2 at the RFB. Nonetheless, the point raised is valid and we investigate the role of mono-SUMOylation in HR-dependent DNA synthesis. As requested by reviewer 3, we have generated a

SUMO-KallR mutated strain (all Lysine being mutated to Arginine) to allow only mono-SUMOylation event to occur. We found that RFB-induced Ura⁺ reversion was increased in this mutant compared to WT (Fig. 5g), supporting that SUMO chain impedes the resumption of HR-dependent DNA synthesis. We added in the discussion part “We noticed that the lack of Pli1 did not increase RDR efficiency whereas preventing SUMO chains does, suggesting that Pli1-dependent mono-SUMOylation events remain necessary to RDR” (page 13-14, lines 430-432)

Minor issues:

6) For the replication slippage assay, what is the frequency of spontaneous replication slippage (i.e. without RFB, as shown in figure 2c, top panel) for all the genetic backgrounds in which this assay was performed. Also, how were the data normalized relative to WT in order to account for a possible difference in the frequency of spontaneous replication slippage among the different mutant backgrounds.

The frequency of replication slippage (spontaneous and upon RFB induction) are now provided in the extended DATA source excel file for each mutant analyzed. To obtain the true occurrence of replication slippage induced by the *RTS1*-RFB, independently of the genetic background, we subtracted the replication slippage frequency of the strain devoid of RFB (considered as spontaneous frequency) from the frequency of the strain containing the *t-ura4sd20<ori* construct, upon expression of Rtf1. This point is now added to the material and method.

7) For Figure 5a, deletion of *pli1* only partly rescued the phenotype. This should be more clearly stated.

This is true and the text is corrected (page 10 lines 309-310).

8) For Figure 5d, the authors stated that the “lack of Pli1 fully rescued the defect in FRB-induced RS in *nup132Δ* cells”. Statistical analysis should be performed to compare WT and *pli1Δ nup132Δ* cells.

The statistics were done and added to Fig 5e ($p < 0.05$ when comparing *pli1Δ nup132Δ* to WT and ns when comparing *pli1Δ nup132Δ* to *pli1Δ*). The text is modified as follows “the lack of Pli1 partly rescued the defect in RFB-induced RS of *nup132Δ* cells (Fig. 5e)” (page 10 lines 315-317). Similarly, statistics were added on Fig 5e to compare *pli1Δ slx8-29* to WT at 32°C (ns) and to Fig 6b to compare Ulp1 overexpressing *nup132Δ* strain to WT (ns).

Reviewer #3 (Remarks to the Author):

(1) Presentation: a more detailed description of the experimental strategy is needed in places. For example: the experimental approach to visualize S-phase specific relocation of the RFB site is described at length, while the genetic replication restart assay is mentioned only very briefly.

A better-detailed explanation of the genetic replication restart assay is now given in the legend of figure 3b, as well as in the material and method (please, see also the response to point 6 of reviewer 2).

(2) The *fml1* data appears to be only very loosely connected to the rest of the paper and disrupts the logical flow. Unless the authors can provide a good explanation why it should be included, I would consider removing it from the current paper.

We considered this comment and removed the data about Fml1.

(3) I take issue with some of the ChIP data in this paper as the authors test exclusively genomic locations that are in immediate vicinity. Can they include additional qPCR data measuring enrichment outside the affected area, to show that the enrichment they observe is specific for the surrounding RFB location.

We added additional ChIP-qPCR data to monitor enrichment at the *ade6* locus outside the RFB locus. These controls were done for ChIP-qPCR against Npp106-GFP (Fig 1e, 2b, 4e, 5c) Man1-GFP (Fig 1e), Sad1-GFP (Fig 1e), Rad51 (Fig 3e, Fig 4c), Ulp1-LexA (Fig 6e) and RPA (Fig 7c). The data obtained support enrichment being specific for the surrounding RFB location. We hope that these data are convincing enough.

(4) Overall, I think the Npp106 ChIP data is an essential addition to the fluorescence microscopy data. It needs to be added for several key strains such as *nup132* and the *nup132 pli1* double mutant.

We performed ChIP-qPCR against Npp106-GFP in the following strains: *pku70Δ* (Fig 2b), *exo1Δ* (Fig 2b), *rad50Δ* (Fig 2b), *rad51-III3A* (Fig 2b), *nup132Δ* (Fig 4e), *nup131Δ nup132Δ* (Fig 4e), *pli1Δ* (Fig 5c), and *pli1Δ nup132Δ* (Fig 5c). The data obtained reinforce the conclusions drawn from the microscopy experiments, excepted in the *pku70Δ* strain, where this additional experiment allowed us to revise our conclusion (please, see the response to point 1 of reviewer 2). We think that these data strengthen our conclusion.

(5) In their study the authors have reached a good understanding of the *nup132* mutant phenotype. However, the phenotype of the *nup132 nup131* double mutant is much less understood. Can the authors provide further insights on this mutant? Can they really interpret its phenotype in the context of a recruitment defect or is it possible (likely?) that this mutant leads to a defect in NPC assembly and leads to for example NPC clustering?

We agree that our interpretation that Nup131 and Nup132 act redundantly to anchor arrested forks to NPCs was an over interpretation of the data. We toned our conclusion down and added the following sentence: "Thus, Nup132 is dispensable to anchor remodeled forks to NPCs. However, the absence of both nucleoporins may modify the NPC structure, making it inefficient for anchoring" (page 9 lines 287-289). However, we have no evidences to support a defect in NPC assembly in the absence of both nucleoporins. We observed only few Npp106-GFP *nup131Δ nup132Δ* cells exhibiting NPC clustering upon growth in rich media, in agreement with the modest NPC clustering phenotype previously scored in this same mutant (Bai et al, MCB 2004).

We uncovered interesting phenotypes in the *nup131Δ nup132Δ* double mutant such as a lack of fork-integrity. However, at this stage, we are unable to link directly those phenotypes with the lack of anchorage to NPCs. Therefore, we decided to not include these data in this current revised version of the manuscript. The focus of this paper is to establish how the NPC is mechanistically engaged at arrested forks to ensure efficient replication restart. So far, the outcome of relocation of damage is often deduced from the phenotypes arising from ablation of anchorage sites at NPCs. One strength of our work, and its novelty, is to genetically separate the role of the NPC in anchorage from its role in promoting DNA repair/replication, thanks to the use of the *nup132* mutant. We hope that our inability to provide further insights on the *nup131Δ nup132Δ* double mutant will not lowered the novelty of our finding.

(6) I am not convinced the authors can conclude that it is specifically SUMO chains that cause toxicity in the absence of Nup132. In order to make this conclusion a SUMO KR all (non-SUMOylatable) mutant would be required.

We generated an endogenous SUMO mutated form in which all Lysine are mutated to Arginine. This SUMO mutated is non-SUMOylatable and generates only mono-SUMOylation events without the formation of SUMO chains, as reported by Prudden et al. (MCB 2009). This mutant resulted in a lack of relocation of the active RFB to the nuclear periphery (Fig 5f), and rescued as well the RFB-induced RS defect of the *nup132Δ* mutant (Fig 5g). We believe that these data reinforce our conclusion that SUMO chains are necessary for relocation but impede HR-mediated DNA synthesis and become toxic to RDR when they cannot be properly processed (such as in the *nup132Δ* mutant).

(7) The HU/MMS sensitivity of *nup132* mutants cannot be rescued by the *pnmt81-ulp1* strains. Is this the *pnmt81-ulp1* condition well understood? Perhaps the authors should concentrate fully on the *lexA-Ulp1* strain in the context of their RFB system, but present a full analysis here including ChIP data.

We considered this comment and removed the data about Ulp1 overexpression on cell sensitivity to replication stress. We performed ChIP-qPCR against Ulp1-*lexA* to demonstrate the tethering to the RFB. We hope that the novel data are now convincing enough.

REVIEWERS' COMMENTS

Reviewer #2 (Remarks to the Author):

The revised manuscript of Kramaz et al is greatly improved. In particular, the new Npp106-GFP ChIP data support the microscopy data in terms of addressing the binding of the active RFB to the NPCs in various mutant backgrounds. Moreover, the conclusions presented in this revised manuscript better reflect the experimental data. Finally, the writing and the flow of the manuscript have also been greatly refined. The revised manuscript is now easier to read and better highlights the novelty of the findings.

Reviewer #3 (Remarks to the Author):

The revised version of this manuscript by Sarah Lambert and colleagues addresses all points, which I raised in my previous critique. I find the manuscript strongly improved and congratulate the authors to an excellent study, which provides new insights into the spatial organization and mechanism of replication restart.

REVIEWERS' COMMENTS

Reviewer #2 (Remarks to the Author):

The revised manuscript of Kramaz et al is greatly improved. In particular, the new Npp106-GFP ChIP data support the microscopy data in terms of addressing the binding of the active RFB to the NPCs in various mutant backgrounds. Moreover, the conclusions presented in this revised manuscript better reflect the experimental data. Finally, the writing and the flow of the manuscript have also been greatly refined. The revised manuscript is now easier to read and better highlights the novelty of the findings.

Reviewer #3 (Remarks to the Author):

The revised version of this manuscript by Sarah Lambert and colleagues addresses all points, which I raised in my previous critique. I find the manuscript strongly improved and congratulate the authors to an excellent study, which provides new insights into the spatial organization and mechanism of replication restart.

We would like to thank the two reviewers for their positive comments and constructive criticisms during this review process.